

# Arctic supraglacial lake derived bathymetry combining ICESat-2 and spectral stratification of satellite imagery

Jinhao Lv [1], Chunchun Gao [1, 2], Chao Qi [1, 2, *], Shaoyu Li [1, *], Dianpeng Su [1, 2], Kai Zhang [1, 2], and Fanlin Yang [1, 2]

[1] College of Geodesy and Geomatics, Shandong University of Science and Technology, 266590 Qingdao, P.R. China
[2] Key Laboratory of Ocean Geomatics, Ministry of Natural Resources of China, 266590 Qingdao, P.R. China

*Correspondence to*: Chao Qi (qichoice007@sdust.edu.cn) and Shaoyu Li (202181020007@sdust.edu.cn)

**Abstract.** Arctic supraglacial lakes volume changes serve as critical indicators of global temperature fluctuations. Accurate lake depth measurements are essential for reliable volume estimation, yet traditional bathymetry methods (e.g., airborne

LiDAR and shipborne sonar) face significant challenges and high costs in the harsh Arctic environment. This study introduces a novel approach using ICESat-2 (Ice, Cloud, and Land Elevation Satellite-2) and Sentinel-2 data to derive supraglacial lake bathymetry. By considering the varying reflectance characteristics across different spectral bands in the water column, we conduct a satellite-derived bathymetry (SDB) method based on spectral stratification using the Otsu algorithm (maximum between-class variance method). Integrating the spectral stratification method with the classical log-transformed linear

regression model (Lyzenga model), we perform accurate bathymetric inversion on multispectral satellite imagery. To verify the effectiveness of the proposed method, we apply it to four representative lakes on the Greenland Ice Sheet (GrIS), using ArcticDEM (Arctic Digital Elevation Model) as reference data. Experimental results demonstrate improved accuracy compared to the classical Lyzenga model, with reductions in root mean square error (RMSE) and mean absolute error (MAE) by up to 13.0% and 14.0%, decreasing from 0.54 m to 0.47 m and from 0.43 m to 0.37 m, respectively. The enhanced accuracy

and scalability of our approach improve the ability to monitor large-scale volume changes in Arctic supraglacial lakes, providing valuable insights into their response to climate change.

## 1 Introduction

The Arctic plays a crucial role in maintaining Earth's temperature balance and exhibits heightened sensitivity to global climate change (Box et al., 2019; Sand et al., 2016; Schmale et al., 2021). The accelerated melting of Arctic glaciers in recent years,

driven by global warming, has exerted substantial negative impacts on the global ecological environment (Box et al., 2022; Lüthje et al., 2006). Greenland Supraglacial Lakes (SGLs) are formed by depressions on the surface of the Greenland Ice Sheet (GrIS). Their volume is influenced by factors such as runoff (meltwater, rain, and refreezing) and lake drainage (Leeson et al., 2015). These changes in water volume within Arctic SGLs are closely linked to ice sheet melting, offering valuable insights into regional temperature variations and the ice sheet's response to different climatic factors.(Beckmann and Winkelmann,

2023). Accurately estimating the volume of these lakes requires detailed bathymetry data, which is particularly challenging to



obtain due to the harsh climatic conditions in the Arctic. Although conventional bathymetric methods, including bathymetric airborne lidar and shipborne sonar, have achieved high levels of maturity and accuracy, their use in polar regions, particularly over ice sheets, remains challenging and expensive (Li et al., 2022; Qi et al., 2022; Qi et al., 2024). This limitation significantly restricts the acquisition of continuous spatiotemporal volume data for SGLs. With advancements in satellite remote sensing

technology, satellite-derived bathymetry (SDB) has emerged as a promising method for estimating bathymetry in clear water areas using multispectral imageries (Ma et al., 2020). Classic bathymetry inversion models, such as the log-transformation linear regression model (Lyzenga model) and the log-transformation ratio model (Stumpf model) (Lyzenga, 1978, 1985; Stumpf et al., 2003), are widely used in water depth inversion, but their accuracy is constrained by the lack of *in-situ* depth data.

In recent years, the launch of spaceborne single-photon altimetry satellite ICESat-2 (Ice, Cloud, and Land Elevation Satellite-2) has partially mitigated challenges in obtaining precise measurement data (Albright and Glennie, 2020; Li et al., 2023). Numerous studies have integrated multispectral technology with ICESat-2 to conduct bathymetric detection and inversion, leveraging both active and passive remote sensing methods. These studies have yielded significant results, primarily in island reef areas far from the mainland. For the bathymetry inversion in island reef areas. Cao et al. (2016) developed a high-precision

bathymetry model for Ganquan Island in the South China Sea by using laser satellite data and optical imagery. This approach leverages active and passive remote sensing techniques, tailored to the specific characteristics and requirements of shallow water bathymetry. Ma et al. (2020) used ICESat-2 data and Sentinel-2 data to retrieve the bathymetry information of the Xisha Islands and Aklin Island in the South China Sea. Thomas et al. (2021) also used Sentinel-2 and ICESat-2 data, combined with the Lyzenga model, Stumpf model, and support vector machine model, to derive bathymetry maps of Florida, Crete, and

Bermuda, Zhao et al. (2024) proposed a seafloor substrate in coral reef areas into sandy and coral, based on the Depth Invariant Index (DII) and conduct bathymetry inversion based an adaptive log-ratio model. In addition, Chu et al. (2023) considered the penetration limit bathymetry in different bands of multispectral imagery and proposed a satellite-derived bathymetry method based on spectral stratification, which was successfully applied to the long line reefs in the Nansha area of China and Buck Island in the United States Virgin Islands, improving the inversion accuracy to a certain extent. For the bathymetry inversion

in Arctic regions, Lin et al. (2012) used multibeam bathymetric data and Landsat TM data to invert the bathymetry of lakes in the Arctic Alaska Coastal Plain. Moussavi et al. (2016) utilized the stereoscopic imaging capability of Worldview-2 data to estimate and validate the bathymetry of SGLs of the GrIS, achieving high accuracy. Melling et al. (2024) used Sentinel-2 data to construct Radiative Transfer Equations (RTE) for different bands and validated them by combining ICESat-2 and ArcticDEM (Arctic Digital Elevation Model) data. Fricker et al. (2021) introduced ICESat-2 data to estimate the meltwater

depth of the Antarctic ice sheet and Greenland, providing a reference for polar water depth inversion. Datta and Wouters (2021) proposed the Watta algorithm, which focused on studying the drainage situation of arctic lakes by utilizing ICESat-2 data and multispectral data. Lv et al. (2024) used the Stumpf model, combined with ICESat-2 and Sentinel-2 imagery, to invert the bathymetry of some SGLs on the GrIS from 2019 to 2023.



The method of SDB using a combination of active and passive remote sensing has been widely applied in the open ocean and
island reefs, but it has been rarely utilized for inverting the bathymetry of polar SGLs. Moreover, the traditional SDB models
(e.g., Lyzenga model and Stumpf model) applied to polar SGL generally do not consider the reflectance diversities of the water
column across different bands (i.e., red band, green band, blue band, and near-infrared band), which limits their accuracy.
Inspired by the spectral stratification bathymetric inversion method applied by Chu et al. (2023) on offshore islands and reefs,
we propose a new SDB method for Arctic SGLs based on spectral stratification, aiming to improve the accuracy of active-
passive remote sensing bathymetric inversion models for Arctic SGL depths. This method divides multispectral imagery into
red, green, blue, and near-infrared layers using the Otsu algorithm for constructing a spectral stratified inversion model. The
Lyzenga model is then applied, using ICESat-2 lake bottom photons as training samples, to derive bathymetry for each band.
The results are validated against high-resolution ArcticDEM data. This study provides a new solution for accurately monitoring
the Arctic SGL volumes and offers effective technical support for predicting Arctic glacier melt and global climate change.
The manuscript is organized as follows: Section 2 describes the study areas and data sources. Section 3 explains the proposed
method. Section 4 presents the experimental results. Section 5 discusses the findings in detail. Finally, the conclusion
summarizes the manuscript.

## 2 Study area and data

### 2.1 Study area

The study area is located in the southwest of the GrIS in the Arctic, the second-largest ice sheet in the world, surpassed only
by the Antarctic Ice Sheet. However, the GrIS is more fragile and sensitive to temperature changes than the Antarctic Ice Sheet
(Robinson et al., 2012). This region contains numerous shallow SGLs with clear water, providing a good condition for
bathymetry retrieval using ICESat-2 laser altimetry and multispectral remote sensing imagery (Feng et al., 2024; Lv et al.,
2024). Four Arctic SGLs are selected for research and analysis, as depicted in Figure. 1. For convenience, these lakes are
referred to as Lake A, Lake B, Lake C, and Lake D. The study aimed to verify the feasibility of the bathymetric inversion
model using these four lakes. Subsequently, the bathymetry data obtained by the proposed method was used to calculate the
volume information of these four lakes.



**Figure. 1 Overview of the study area and typical lakes. The background consists of Sentinel-2 multispectral imagery depicting the study area and the four lakes. Yellow, red, blue, and green lines represent ICESat-2 tracks traversing these four lakes, respectively. The black points indicate ICESat-2 raw photon data.**

## 2.2 Study data

### 2.2.1 Sentinel-2 multispectral imagery

The Sentinel-2 satellite, a medium-resolution multispectral satellite launched by the European Space Agency (ESA), is a critical component of its Earth observation mission. The Sentinel-2 system comprises three satellites (i.e., Sentinel-2A,



Sentinel-2B, and Sentinel-2C) which were launched on June 23, 2015, March 7, 2017, and September 5, 2024, respectively (Ma et al., 2020). The Sentinel-2 satellite features 13 bands covering visible light, near-infrared, and shortwave infrared bands, with some bands having a resolution of up to 10 m (Hedley et al., 2018). Sentinel-2 imagery spans a width of up to 290 km

and a single satellite revisit period of 10 days. In this study, Sentinel-2 imagery of lakes A, B, C, and D investigated was obtained on July 4, 2020, July 4, 2020, July 17, 2022, and July 15, 2021, respectively. The Sentinel-2 multispectral imagery can be downloaded for free from the Internet (https://dataspace.copernicus.eu/explore-data/data-collections/sentinel-data/sentinel-2).

### 2.2.2 ICESat-2 single-photon LiDAR data

The ICESat-2 satellite orbits at an altitude of approximately 500 km with an inclination of 92°, observing the Earth's surface between latitudes 88°S and 88°N. The platform is equipped with the Advanced Topographic Laser Altimeter System (ATLAS) single photon lidar and auxiliary system, which determines the distance between the spacecraft and the Earth's surface by measuring the round-trip time of photons (Markus et al., 2017). The ICESat-2/ATLAS laser emits laser pulses with a wavelength of 532 nm and a width of 1.5 ns at a frequency of 10 kHz, forming overlapping light spots along the Earth's surface

with an orbital spacing of approximately 0.7 m (Magruder et al., 2021). The left and right points of each pair of beams are approximately 90 m apart in the transverse track direction and approximately 2.5 km apart along the track direction. Paired tracks are approximately 3.3 km apart in the transverse track direction (Neumann et al., 2021). The ICESat-2 data acquisition dates used in this study were July 6, 2020, July 6, 2020, July 14, 2022, and July 15, 2021, respectively. The ICESat-2 data can be obtained freely from the Internet (https://search.earthdata.nasa.gov/). This study utilized ICESat-2 data intercepted from

Sentinel-2 data as training data and assumed a lake water depth of 0 m at the edges of the ICESat-2 tracks. Please note that this study assumes that the depth at the intersection of the ICESat-2 track and the lake's land-water boundary is 0 m.

### 2.2.3 ArcticDEM data

ArcticDEM can be obtained for free from the Polar Geospatial Center at the University of Minnesota in the United States (https://www.pgc.umn.edu/data/arcticdem/), and has the characteristics of a large coverage area and high spatial resolution. It

covers all land areas above 60° north latitude, with a spatial resolution of up to 2 m, and has significant reference value for topographic research in the Arctic region (Melling et al., 2024). The high-resolution Arctic digital elevation model ArcticDEM data is extracted and output as strip-shaped DEM data by SETSM (Surface Extraction with TIN based Search space Minimization), preserving the time information of the original data and allowing users to service data for research and analysis as needed (Morin et al., 2016). It should be pointed out that the ArcticDEM data used in this study were obtained on May 11,

2020, April 20, 2022, and March 12, 2021.



**Table 1: Acquisition dates of the data used in the study**

| Study Area | | Multispectral Imagery (Sentinel-2) | Training Data (ICESat-2) | Validation Data (ArcticDEM) |
|---|---|---|---|---|
| | Lake A | 04/07/2020 | 06/07/2020 | 11/05/2020 |
| Southwest Greenland Ice-sheet | Lake B | 04/07/2020 | 06/07/2020 | 11/05/2020 |
| | Lake C | 17/07/2022 | 14/07/2022 | 20/04/2022 |
| | Lake D | 15/07/2021 | 15/07/2021 | 12/03/2021 |

## 3 Methodology

To address the challenges and the limitations of traditional bathymetric methods, which do not consider the varying penetration of electromagnetic waves of different wavelengths into water, this study combines single-photon LiDAR ICESat-2 data with multispectral imageries from Sentinel-2. Using the Otsu algorithm (Otsu, 1975), spectral stratification of the multispectral images is performed based on the reflectance differences at various water depths across different bands. The stratified spectral layers are then combined with the ICESat-2 bathymetric data to construct an optimized Lyzenga

model for each spectral layer. The detailed workflow is illustrated in Figure. 2.







<p style="text-align:center"><b>Figure. 2 The workflow of the proposed SDB method.</b></p>

## 3.1 Data processing

### 3.1.1 Pre-processing of multispectral imagery

The Sentinel-2 data provided by the ESA is divided into L1C and L2A levels. The L1C level data product is a geometric
precision correction radiographic product that has not undergone radiometric correction. L2A products are products that
undergo radiation correction processing based on L1C. L1C radiation correction can be processed using the Sen2Cor plugin




provided by the ESA to convert L1C level data into L2A level data for radiation correction. The water column was extracted
from multispectral imagery using water-land separation methods, i.e., the Normalized Difference Water Index (NDWI) (Eq.

(1)) and threshold-based grayscale segmentation (McFeeters, 1996).

$$NDWI = \frac{Green - NIR}{Green + NIR} \tag{1}$$

where *Green* represents the reflectance at the green band, and *NIR* represents the reflectance at the near-infrared band.
Since the multispectral imagery obtained contained partially unmelted ice cover on the acquisition date, a mask was applied
to the ice cover on the lake to ensure the accuracy of the study during the water column extraction process.

**3.1.2 ICESat-2 bathymetric photons processing**

The ICESat-2 single photon data is subject to a large amount of noise information in the solar background imagery, so noise
photon removal processing is required. In this study, we used the density based spatial clustering algorithm DBSCAN (Density
Based Spatial Clustering of Applications with Noise) to segment point cloud data into surface photon data and underwater
photon data, to remove noise and obtain the required underwater photon data (Ma et al., 2020). Figure. 3 shows four ICESat-

2 data tracks used for four lakes. Due to the fact that the ICESat-2 ATL03 data did not consider the deviation of laser
propagation caused by different refractive indices between air and water columns, it was necessary to perform refraction
correction on the underwater photons. Finally, more accurate bathymetric photons were obtained for constructing a bathymetry
inversion model. The refraction correction model used in this study was the Parrish 2019 model (Parrish et al., 2019). Since
the surface of Arctic SGLs is usually calm, the study did not consider the effects of waves and tidal phenomena on photons,

the equation of the refraction correction was expressed as:

$$D = \left\{ 1 - \tan\left[ \frac{(\theta_1 - \theta_2)}{2} \right] \sin(\theta_1 - \theta_2) \right\} R_c \tag{2}$$

where $D$ is the corrected bathymetry, $\theta_1$ is the laser incidence angle, $\theta_2$ is the laser refraction angle, and $R_c$ is the corrected laser
transmission distance derived from Snell's Law.

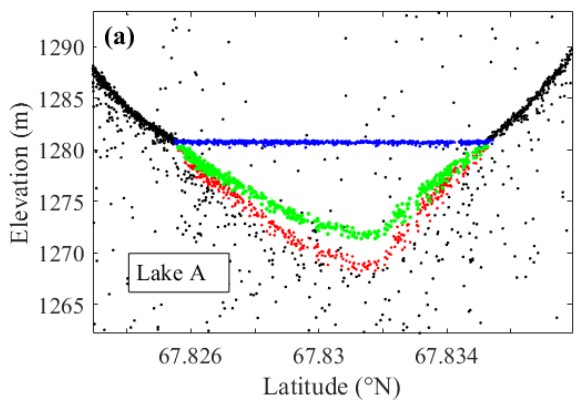
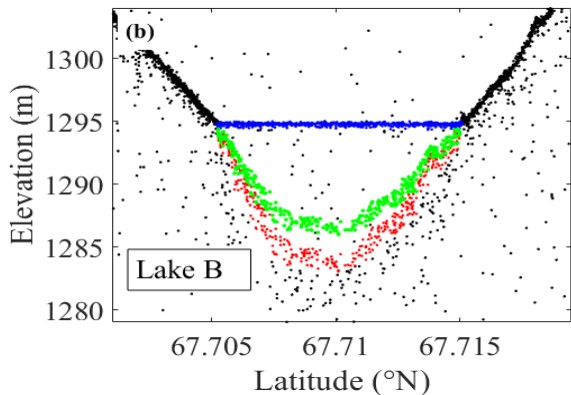



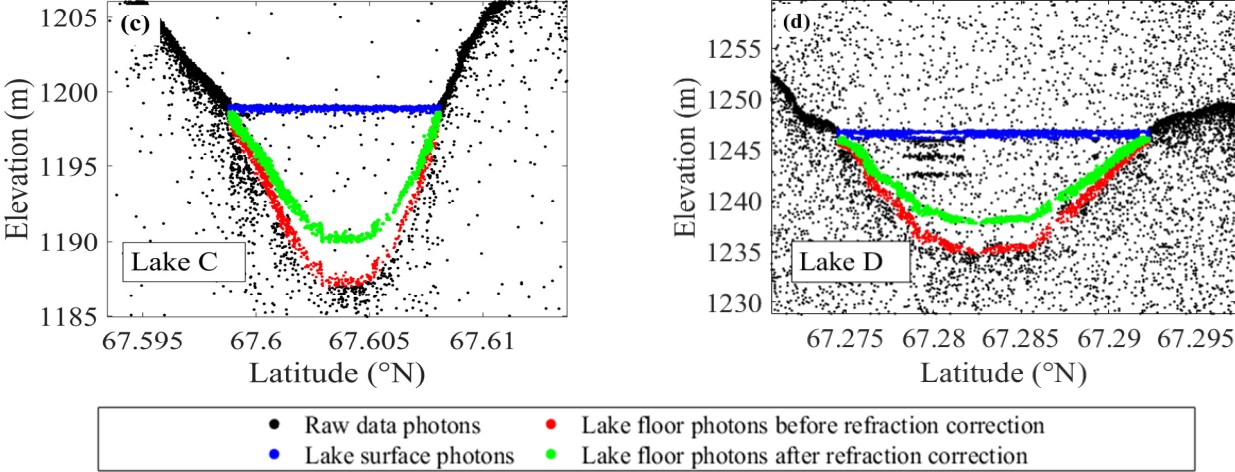

**Figure. 3 Extraction and correction of ICESat-2 bathymetry photons for four lakes. (a) Data track for Lake A on 06/07/2020. (b) Data track for Lake B on 06/07/2020. (c) Data track for Lake C on 14/07/2022. (d) Data track for Lake D on 15/07/2021.The black points represent the ICESat-2 ATL03 raw data photons, the blue points represent the lake surface photons, the red points represent the lake floor photons without refraction correction, and the green points represent the corrected lake floor photons, which can be used to construct the bathymetry inversion model.**

It should be noticed that Arctic SGLs are typically dynamic, with their size and shape exhibiting significant changes over short periods. Therefore, The ICESat-2 and Sentinel-2 data used for Arctic SGL bathymetry inversion should be acquired simultaneously whenever possible. If temporal discrepancies between the data sources result in spatial misalignment of lake features in ICESat-2 photon and Sentinel-2 imagery, a vertical adjustment of ICESat-2 photon can be applied. This adjustment ensures that the depth value of the photon at the lake boundaries is set to 0 m, thereby mitigating systematic errors caused by temporal mismatches between datasets.

## 3.2 Spectral stratification for multispectral imagery based on Otsu algorithm

The spectral stratification method used in this study employed the Otsu algorithm, which does not require input parameters (Otsu, 1975). By utilizing the penetration characteristics and reflectance differences of water across various spectral bands, multispectral images of water stratified into four layers: near-infrared band, red band, blue band, and green band. Specifically, the Otsu algorithm was first applied to determine the water extraction threshold in the near-infrared band layer. Subsequently, water in the near-infrared band was masked, and the same method was used to extract water in the red band. Next, both the near-infrared and red bands were masked to perform threshold segmentation for extracting water in the green band. Finally, by masking the extracted near-infrared, red, and green bands, water in the blue band was obtained, thereby achieving spectral stratification of satellite-derived multispectral images (Figure. 4).



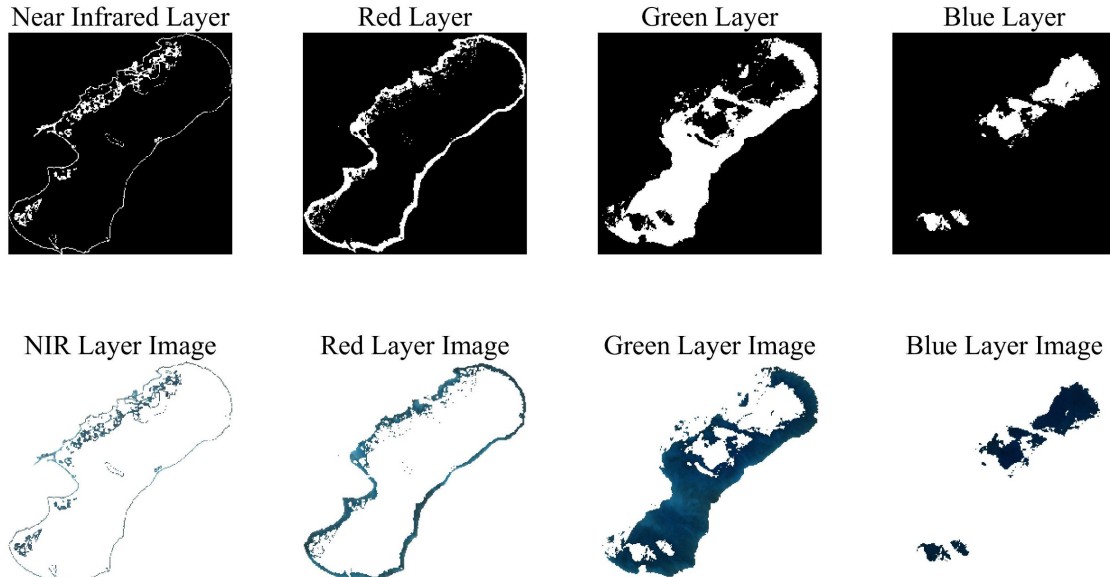

**Figure. 4 Binarized segmentation and spectral stratification of the Lake A**

### 3.3 Spectral stratified bathymetry inversion model

Lyzenga's log-transformed linear regression model links spectral data to bathymetry using the log sum of water reflectance differences from deep-water values (Lyzenga, 1978, 1985). This study constructed a spectral stratified bathymetry inversion model based on the traditional Lyzenga model. By leveraging the varying penetration abilities of electromagnetic waves in water, satellite-derived multispectral images were segmented into different layers, and a bathymetry inversion model was established combining the traditional Lyzenga model. Owing to the relatively small lake surface area of the near-infrared layer and the red layer and the insufficiency of ICESat-2 bathymetry training photons in this study, the near-infrared layer, the red layer, and the green layer were combined for processing. In other words, Arctic SGLs were divided into green and blue layers for bathymetry inversion, the expression was:

$$Z = a_{g0} + \sum_{i=1}^{N} a_{gi} \ln\left[L(\lambda_i) - L_\infty(\lambda_i)\right] \tag{3}$$

$$Z = a_{b0} + \sum_{i=1}^{N} a_{bi} \ln\left[L(\lambda_i) - L_\infty(\lambda_i)\right] \tag{4}$$

where $Z$ was the predicted bathymetry result, $a_{g0}$ and $a_{gi}$ were the corresponding parameters of the green light layer, $a_{b0}$ and $a_{bi}$ were the corresponding parameters of the blue light layer, $L(\lambda_i)$ was the reflectance value of the $i$-th band, and $L(\lambda_\infty)$ was the reflectance of the deep-water zone in the $i$-th band.





## 3.4 Evaluation metrics

The bathymetric information derived from the above two bathymetry inversion models was validated and analysed using the ArcticDEM data as the reference. The coefficient of determination ($R^2$), root mean square error (RMSE), and mean absolute error (MAE) between the two datasets were calculated to quantitatively validate the effectiveness of the proposed method (Hodson, 2022). The formulas for these metrics are as follows:

$$R^2 = \frac{\sum_{i=1}^{n}\left(\hat{h}_i - \bar{h}\right)}{\sum_{i=1}^{n}\left(h_i - \bar{h}\right)} \tag{5}$$

$$RMSE = \sqrt{\frac{\sum_{i=1}^{n}\left(h_i - \hat{h}_i\right)^2}{n}} \tag{6}$$

$$MAE = \frac{1}{n}\sum\left|h_i - \hat{h}_i\right| \tag{7}$$

where $\hat{h}_i$ is the predicted water depth, and $h_i$ is the ArcticDEM-derived validation water depth.

## 4 Result and analysis

Utilizing the datasets detailed in Section 2 and the bathymetry inversion methods outlined in Section 3, this section performed bathymetry inversion for Lakes A, B, C, and D. Additionally, the accuracy of the inversion results was validated and compared using ArcticDEM data. Section 4.1 and Section 4.2 presented both qualitative and quantitative evaluations of the bathymetry inversion results obtained from the Lyzenga model and the spectral stratified model to assess the feasibility and accuracy of the spectral stratified bathymetry inversion model.

## 4.1 Qualitative analysis

To verify the effectiveness of the proposed method, we employed both the traditional Lyzenga model (without spectral stratification) and the spectral stratified method to derive the bathymetric information of Lakes A, B, C, and D. The accuracy of bathymetric inversion was compared and analysed using ArcticDEM as benchmark data.

The Lyzenga model and the spectral stratified model were applied to the four lakes in the study area, and the bathymetry inversion results were shown in Figure. 5. In this study, the bathymetry inversion results of the green and blue layers were combined to determine the overall lake depth. It should be pointed out that the study excluded areas covered by unmelted ice sheets on the lakes. Consequently, Figure. 5 shows some regions without bathymetry data.









**Figure. 5 Lake bathymetry inversion results using the Lyzenga model and the spectral stratified model. (a) Sentinel-2 imagery and bathymetry results for Lake A using the Lyzenga model and the spectral stratified model. (b) Sentinel-2 imagery and bathymetry results for Lake B using the Lyzenga model and the spectral stratified model. (c) Sentinel-2 imagery and bathymetry results for Lake C using the Lyzenga model and the spectral stratified model. (d) Sentinel-2 imagery and bathymetry results for Lake D using the Lyzenga model and the spectral stratified model.**

As shown in Figure. 5, the SDB inversion methods utilizing both active and passive remote sensing can achieve the bathymetry of Arctic SGLs. The bathymetric results inverted by the traditional Lyzeng model and the spectral stratified method were generally consistent; however, discrepancies were observed in certain regions, underscoring the reliability and feasibility of the spectral stratified model. It should be noted that the ArcticDEM data only contains spatial information of the lake bottom, and lacks water surface elevation information when obtaining bathymetry benchmark data. Therefore, in this study, the edge position of the ArcticDEM lake was determined using ICESat-2 data, with this elevation serving as the water surface elevation. Additionally, there was a temporal discrepancy of two to four months between the ArcticDEM data and the Sentinel-2 data in this study. However, the sediment in the lakes of the experimental area primarily consists of bedrock, a type of material that remains stable and does not undergo significant changes over short periods. Consequently, it is feasible to use the ArcticDEM as validation data in this study (Melling et al., 2024).

## 4.2 Quantitative analysis

To quantitatively and visually demonstrate the performance of the method, the validation results of the SDB using the traditional Lyzenga model and the spectrally stratified Lyzenga model are presented in Figure. 6.

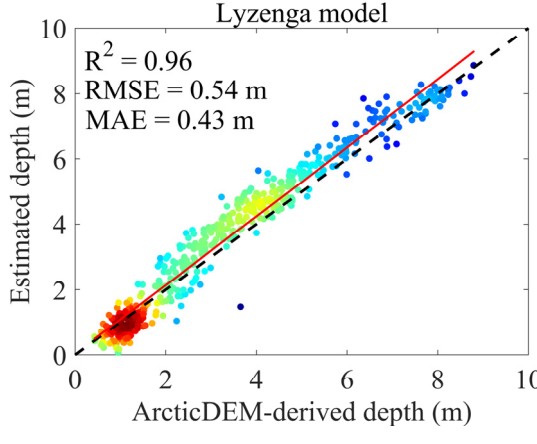

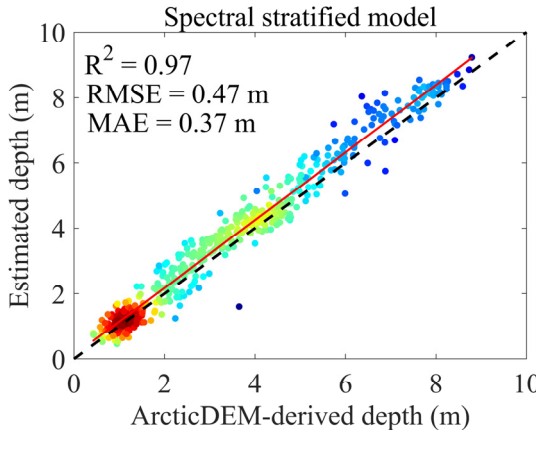

(a)





**Figure. 6 Comparison of bathymetric inversion results using the Lyzenga model and spectral stratified Lyzenga model for four lakes. (a) Lake A bathymetry validation. (b) Lake B bathymetry validation. (c) Lake C bathymetry validation. (d) Lake D bathymetry validation. The point represents the ArcticDEM validation points, the black dashed line indicates the 1:1 line, and the red solid line represents the data fitting line**

245



To visually demonstrate the improvement in the bathymetric inversion accuracy achieved by the spectral stratification method, we selected Lake A as a case study to compare the traditional Lyzenga model with the spectral stratified Lyzenga model. Based on the bathymetric inversion results shown in Figure. 5, the study examines the differences in depth inversion results between the two methods. In areas where relatively obvious discrepancies are observed, ArcticDEM data is used for accuracy validation, as illustrated in Figure. 7.

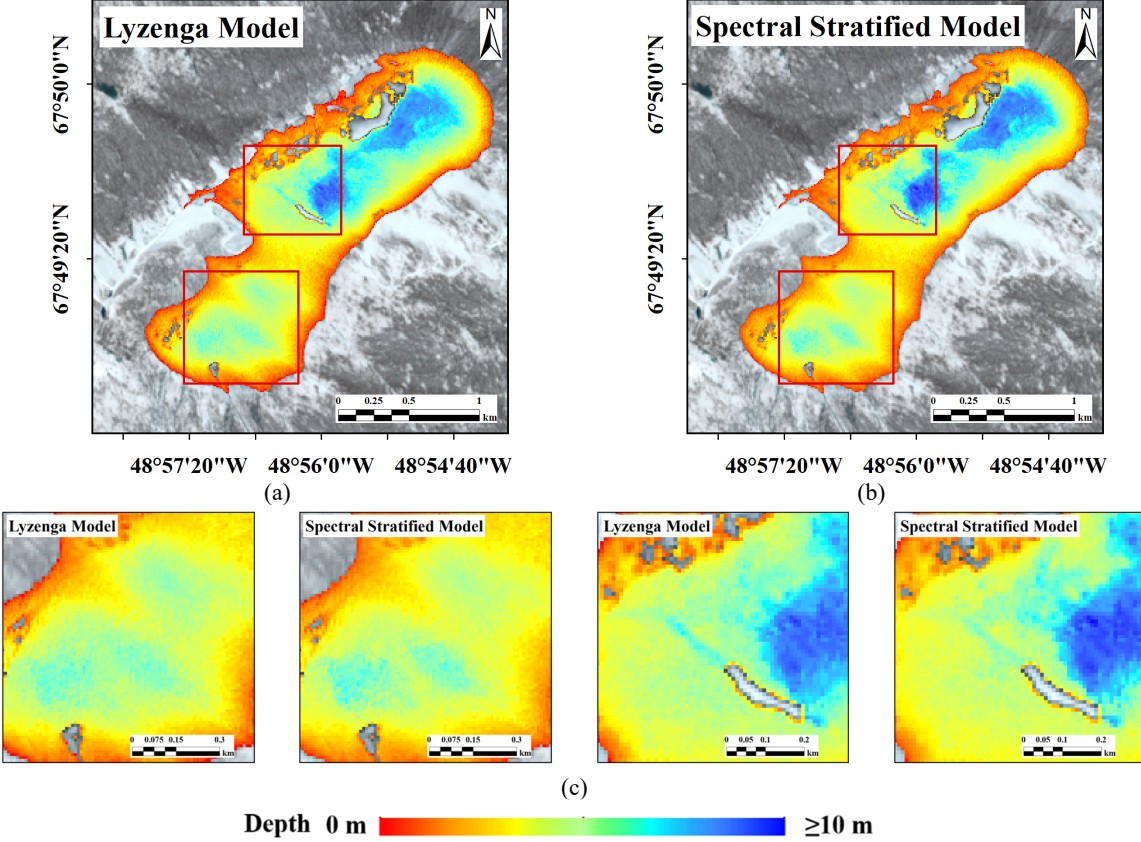

**Figure. 7 Comparison of the bathymetric maps using the Lyzenga model and the spectral stratified model with the Lake A. (a) The depth inversion results derived from the Lyzenga model. (b) The depth inversion results are derived from the spectral stratified model. The red box highlights areas with relatively notable differences in depth inversion between the two models. (c) A detailed view of the water area within the red box.**

A qualitative analysis of the bathymetric inversion results for Lake A indicates that the discrepancies between the Lyzenga model and the spectral stratified model are primarily concentrated in the 2-6 m range, with more pronounced differences at the transitions between different spectral layers.



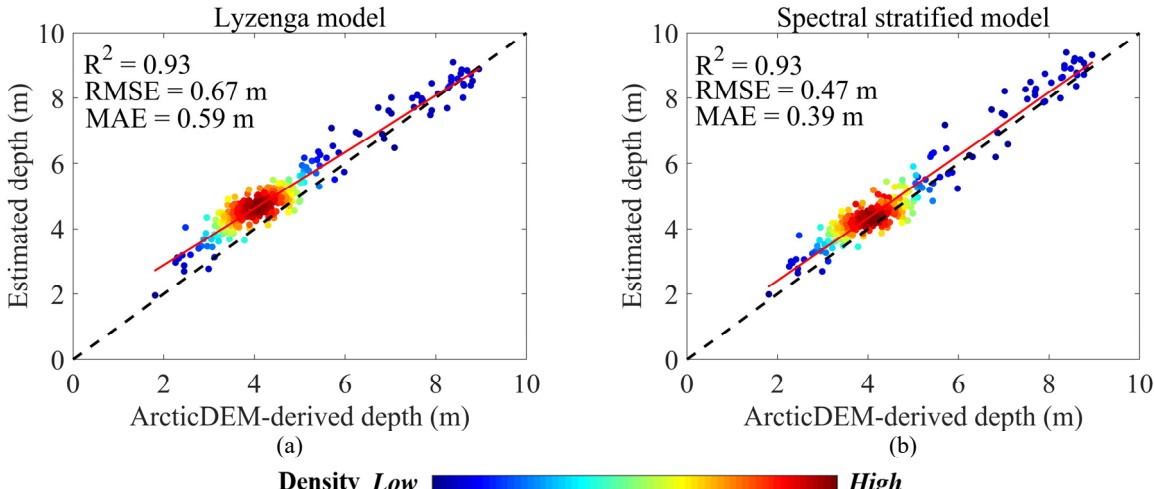

**Figure. 8 Comparison of accuracy using the ArcticDEM data with the Lyzenga model and the spectral stratification model of Lake A. (a) The Lyzenga model. (b) The spectral stratified model. The black dashed line indicates the 1:1 line, and the red solid line represents the data fitting line.**

According to Figure. 6, $R^2$ is relatively high, all of which could reach 0.9 or above, proving that the Lyzenga model is rational for application in these Arctic SGLs. Specifically, the RMSE of the bathymetry inversion for Lake A decreased from 0.54 m to 0.47 m, and the MAE decreased from 0.43 m to 0.37 m, with reductions of 13.0% and 14.0%, respectively. For Lake B, the RMSE and MAE of the spectral stratified bathymetry inversion accuracy values were 0.61 m and 0.46 m, respectively, with decreases of 9.0% and 13.2%. In the case of Lake C, the RMSE reduced to 0.50 m, with reductions of 7.4%. For Lake D, the RMSE reduced to 0.72 m and MAE reduced to 0.52 m, which the RMSE decreased by 5.3% and MAE decreased by 8.8%. Validation results for the red box region of Lake A indicate that the spectral stratified method achieves higher accuracy than the traditional Lyzenga model too, with the RMSE reduced from 0.67 m to 0.47 m, a decrease of 29.9%, and the MAE reduced from 0.59 m to 0.39 m, a decrease of 33.9%, as shown in Figure. 8. For the points with significant absolute errors in the validation results, this may be related to anomalies in pixel reflectance values caused by errors during the radiometric correction of the multispectral imagery data. The validation results further demonstrate the effectiveness of the spectral stratification method in improving bathymetric inversion accuracy. In summary, the experimental results demonstrate that the spectral stratified method utilized in the study effectively improves the accuracy of bathymetry inversion for Arctic SGLs.

## 5 Discussion

### 5.1 Estimation of lake volume

This study extended the application of the spectral stratified method for bathymetry inversion mentioned above, using the spectral stratified model to derive the bathymetry results and fit the surface of the lake bottom. To facilitate the assessment of



volume changes in Arctic SGLs, the volumes of lakes A, B, C, and D were derived through the 3D reconstruction method, as shown in Figure. 9.

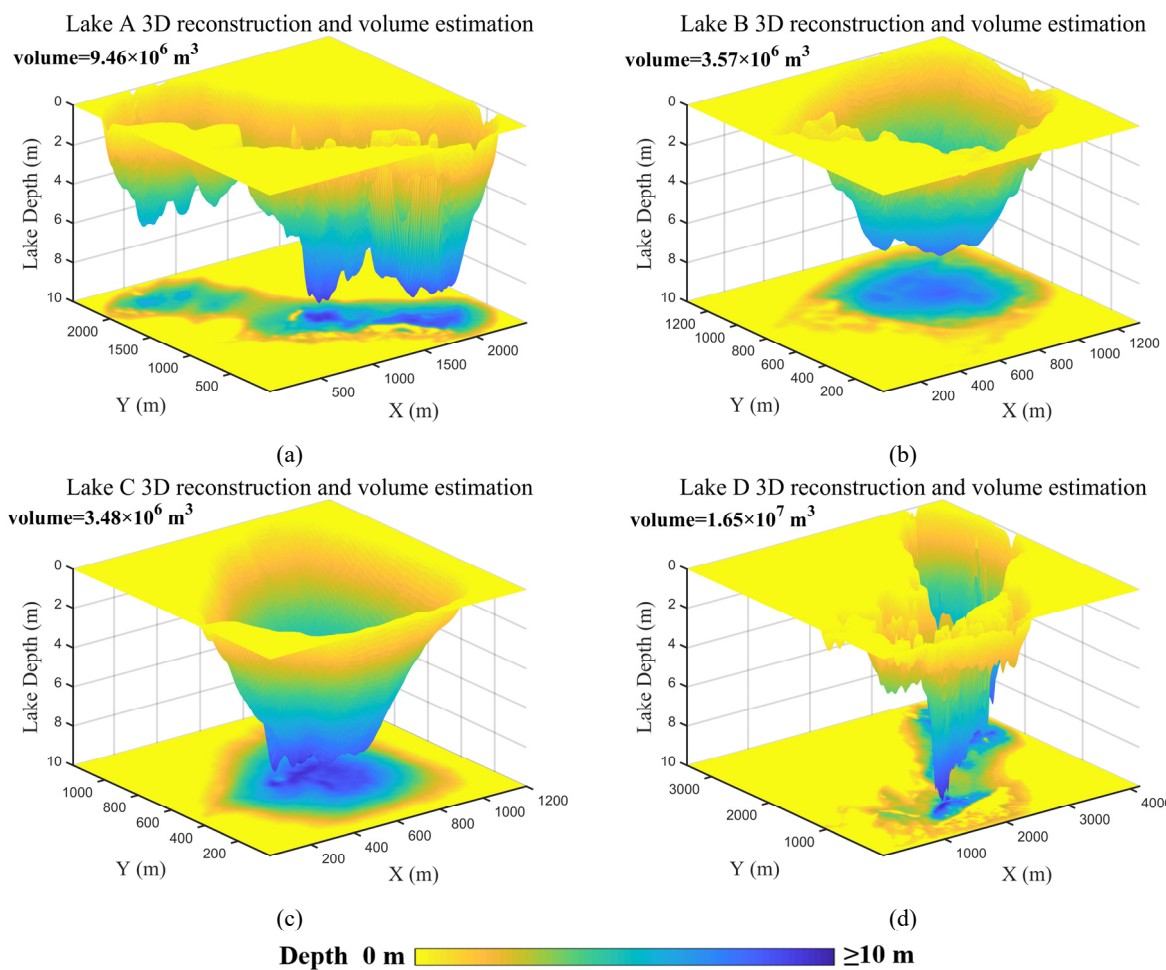

**Figure. 9 Volume estimation of spectral stratification bathymetry and 3D reconstruction of Lakes A, B, C, and D.(a) Lake A 3D reconstruction and volume estimation. (b) Lake B 3D reconstruction and volume estimation. (c) Lake C 3D reconstruction and volume estimation. (d) Lake D 3D reconstruction and volume estimation.**

As shown in Figure. 9, the volumes of Lake A, B, C, and D were $9.46 \times 10^6 \, \text{m}^3$, $3.57 \times 10^6 \, \text{m}^3$, $3.48 \times 10^6 \, \text{m}^3$, and $1.65 \times 10^7 \, \text{m}^3$. Compared to the low temporal resolution of ArcticDEM data, the method of calculating lake volume using bathymetry information obtained by the study approach is more effective and better meets the needs of long-term accurate monitoring of lake volume. It has reference value for studying the volume changes of lakes above arctic glaciers and their relationship with climate-influencing factors.



## 5.2 Challenges and limitations of the spectral stratification method

While the spectral stratification-based method enhances the accuracy of bathymetric inversion compared to traditional
approaches, such as the Lyzenga model, several limitations need attention. First, the method's accuracy is constrained by the
number and distribution of ICESat-2 training samples. Due to the limited amount of strip-shaped data, there were not enough
training samples, which hindered further improvement in model accuracy. Additionally, in some regions of shallow Arctic
SGLs, where bathymetric variations are minimal, the impact of spectral penetration differences is limited, resulting in only
marginal improvements in accuracy. Secondly, the dynamic nature of Arctic SGLs, which experience significant
morphological changes over short periods, poses a challenge when combining ICESat-2 and Sentinel-2 data for bathymetric
inversion. To achieve accurate results, the temporal synchronization of these data sets is critical, which places higher demands
on data acquisition and availability. As a result, there is a limited pool of suitable ICESat-2 and Sentinel-2 data for conducting
effective spectral bathymetric inversion in these lakes.

## 6 Conclusion

The study employed the spectral stratification method and the Otsu algorithm to determine reflectance thresholds across
different bands, segmenting the lake's multispectral satellite imagery into distinct spectral layers. Subsequently, both the
traditional Lyzenga model and the spectral stratified Lyzenga model were utilized to invert the lake's bathymetry, with
ArcticDEM used for accuracy validation. The main conclusions are as follows:

(1) For cases with minimal spatial discrepancies between ICESat-2 lake bottom photons and multispectral imagery, vertical
adjustments can be made to the ICESat-2 photons so that corresponding photons at the lake's edge indicate a depth of 0 m.

(2) Integrating the spectral stratification algorithm into the SDB method improves inversion accuracy, with the most significant
improvement observed for Lake A, where RMSE and MAE were reduced by 13.0% and 14.0%, respectively.

(3) The spectral stratified SDB method requires high-quality data, with limited datasets suitable for SDB. For small, shallow
polar SGLs, the optimization of inversion accuracy through spectral stratification is not significant, limiting improvements in
inversion precision. Despite some limitations, the spectral stratified SDB method can effectively enhance the accuracy of SDB
for polar SGLs, providing more timely and precise depth estimates. This approach offers valuable data support for studies of
polar SGLs where in-situ depth measurements are challenging and serves as a reference for research on environmental changes
in polar regions.



*Code and data availability.* All datasets used in this study are freely available from the sites mentioned in Section 2.2, and the
code can be obtained upon request from the corresponding author.

*Author contributions.* JL contributed to methodology development and drafting the original manuscript. CQ contributed to the
design and conceptualization. CG, CQ, SL, DS, and FY contributed to reviewing and editing the manuscript.

*Competing interests.* The authors declare no conflict of interest.

*Acknowledgements.* This work was supported by the National Natural Science Foundation of China (42304051), the National
Key R&D Program of China (2023YFB3907204), the Shandong Provincial Natural Science Foundation (ZR2024QD062), and
the Young Taishan Scholar Project of Shandong Province (tsqnz20230617). We sincerely thank the NASA, ESA, and Polar
Geographic Data Center for providing data.

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
