# Peer review of "Arctic supraglacial lake derived bathymetry combining ICESat-2 and spectral stratification of satellite imagery"

_EGUsphere, 2025_

## Referee Comment (RC2)

**Arctic supraglacial lake derived bathymetry combining ICESat-2 and spectral stratification of satellite imagery**

Jinhao Lv et al.

Review by Ian Willis

Intro

This is quite an interesting, small-scale study – deriving bathymetries of 4 lakes on the GrIS using a combination of Sentinel-2 and ICESat-2 data. The paper compares what the authors refer to as a "classic bathymetry inversion model", notably the "log-transformation linear regression model (Lyzenga model)" with a modified version of it, which calculates depths separately for different spectral bands and then combines the results to produce an overall bathymetry. It appears as though the ICESat-2 data are used to calibrate parameters in the 'classic' and 'stratified' versions of the Lyzenga model, and then the two bathymetries for the 4 lakes are compared (validated) against ArcticDEM bathymetries (strips collected 2-4 months before the Sentinel data). The 'stratified' version of the Lyzenga model does a modestly better job at capturing bathymetries, as measured using $R^2$, RMSE and MAE, when compared against the ArcticDEM data.

I have a few main issues with the paper and recommendations for improving it, and a long list of small line by line comments, which I suggest would need to be addressed too before the paper is ready to be published.

Main Comments

1. The paper does not adequately acknowledge all the work that has been done by many people over the last decade +, which uses a physically-based algorithm to calculate water depths on the GrIS or Antarctic ice shelves from optical imagery (MODIS, Landsat, ASTER, Sentinel-2). For example, none of the following papers is mentioned or acknowledged in this context.

**Philpot, W.D. 1989.** - Bathymetric mapping with passive multispectral imagery. Appl. Opt., 28(8), 1569–1578. Original algorithm.

**Sneed, W.A. & Hamilton, G.S. (2007)** – First time algorithm used on ice mass using ASTER to estimate supraglacial lake depth/volume on the Greenland Ice Sheet. AGU Publicationsdigitalcommons.library.umaine.edu

**Sneed, W.A. & Hamilton, G.S. (2011)** – Validation of the method with satellite imagery over glacial melt ponds (Greenland), confirming the approach's accuracy. Cambridge University Press & Assessment+1

**Georgiou, S. et al. (2009)** – Applies and extends the Sneed & Hamilton method to track seasonal evolution of a West Greenland supraglacial lake with ASTER. Cambridge University Press & Assessment+1

**Banwell, A.F. et al. (2014)** – Uses the Sneed & Hamilton method to estimate supraglacial lake depths on **Larsen B Ice Shelf (Antarctica)** and in Paakitsoq, Greenland. Aberystwyth Universityrepository.cam.ac.uk

**Pope, A. et al. (2016)** – Implements/assesses the radiative-transfer (Sneed & Hamilton–type) depth retrieval with **Landsat-8** in West Greenland and compares with other multispectral methods. Copernicus TC+1

**Moussavi, M.S. et al. (2016)** – Derives and validates supraglacial lake volumes over West Greenland using **WorldView-2** with the RTE/Sneed-Hamilton framework. ScienceDirect+1NOAA Institutional Repository

**Williamson, A.G. et al. (2017)** – "FASTER" workflow applies the Sneed & Hamilton physically based algorithm to MODIS for automated lake area/volume tracking on the Greenland Ice Sheet. ScienceDirect+1

**Williamson, A.G. et al. (2018)** – Dual-satellite (Sentinel-2 & Landsat-8) analysis over Greenland; depth estimates follow the Sneed-&-Hamilton-style RTE with recommended parameters. Copernicus TCstatic.cambridge.org

**Moussavi, M.S. et al. (2020)** – Detects **Antarctic ice-shelf** lakes (Amery, Roi Baudouin, Nivlisen, Riiser-Larsen) and computes depths/volumes from **Landsat-8/Sentinel-2** using the Sneed & Hamilton-derived approach. MDPI

**Dell, R., et al (2020).** Lateral meltwater transfer across an Antarctic ice shelf. *The Cryosphere, 14*(7), 2313–2330. [https://doi.org/10.5194/tc-14-2313-2020](https://doi.org/10.5194/tc-14-2313-2020). Uses the algorithm to calculate water depths from Landsat imagery across an Antarctic ice shelf.

The paper below is mentioned, but not explicitly in the context of using the 'Philpot' approach:

**Melling, L. et al. (2024)** – Intercompares RTE (using literature coefficients from Sneed & Hamilton and others) with ICESat-2/ArcticDEM across lakes in **southwest Greenland**. Copernicus TC+

The well used 'Philpot' approach should be acknowledged and it would also be useful to mention (or show) how the 'Philpot' equation varies from the classic and modified Lyzenga models. Of course, it would also be valuable to actually use the 'Philpot' equation (e.g. with the parameters advocated by Pope et al 2016) to compare against the 2 calibrated versions of the Lyzenga model and the ArcticDEM.

2.  The paper remains rather limited in scope. In essence it has applied two versions of an empirical model: one, the 'classic' Lyzenga model, that has already been applied to a GrIS lake by Lv et al 2024; and a modified version of it that uses

different spectral bands, that has so far only been applied outside of the context of GrIS lakes. The models were calibrated using ICESat data and validated against ArcticDEM data. There appears to be quite marginal improvement in using the modified version of the model. It is a methods paper, and it doesn't tell us anything about the glaciology or hydrology of GrIS surface lakes. It would be a more compelling paper if it had gone further.

It could have remained methods based but compared also the 'Stumpf' model and the 'Philpot' model for the 4 lakes. It could have told us how transferable the model is – it appears as though the model was calibrated separately for each lake? What were the different parameter values associated with these calibrations? What happens if you apply the model calibrated for one lake to the other lakes? What errors occur? How universal is the model? What happens if you combine all the ICESat-2 data with all the corresponding spectral data to produce an 'all lake' model? How transferable is that? The model needs testing against data that are outside of the lakes that were used for calibration.

It could also have been expanded to tell us about the hydrology of the lakes by calculating their volumes over summer seasons and between years.

3. The work uses the NDWI (which uses the Green and NIR bands) to separate water from ice. This is unusual as virtually all the papers I know about (see e.g., those listed above) use the variant of NDWI tailored for glacier ice and water separation that uses the blue and red bands instead of green and NIR:

$$NDWIice = (Blue-Red)/(Blue+Red)$$

Did the authors consider using this? Why did they opt for the Green and NIR bands? Could we be shown whether it makes a difference to the results?

4. There is confusion throughout the paper about the definition of what the authors refer to as "a satellite-derived bathymetry (SDB) method". See lines 12-15 where it is first mentioned and defined, but then all subsequent references to it as well. Need to clarify that this is the *complete* method that is being used in this paper, **i.e. the Lyzenga model applied to optical satellite data (Sentinel-2), calibrated using altimetry data (ICESat-2).** And the paper uses two versions of the Lyzenga equation. It is not just the use of optical data alone as is sometimes implied in the manuscript.

5. The methodology is not clearly articulated in a consistent way throughout the paper, and parts of the methods appear in the results (see line by line comments below). A key aspect of the methodology that is not articulated clearly up front is that the ICESat-2 data are being used to calibrate the parameters of the two versions of the Lyzenga model. The equation for the 'traditional' version of the Lyzenga model and how it is applied are not explained. Does the traditional version use just one waveband from the optical data? If so which one?

Line by line comments

15 'verify' => 'validate' [as 'validate' is used subsequently in the paper].

17. say you're using 'time stamped' DEM strips here.

19-21. Do you need a sentence before this one explaining what you did to enable you to make this statement? How did you prove it is 'scalable' and what do you mean by that? I'm not convinced you've proved it's scalable, have you?

26. Is Luthje et al a good reference here? Does it explicitly consider 'ecological' issues?

28-30. Beckmann and Winkelmann, 2023 do not consider lakes in their paper as implied. Clarify what you mean by this sentence and use an appropriate ref.

32. **You can't sail ships on ice sheet surface lakes so delete this!** Your 3 refs all relate to lidar. Could you quote studies using bathymetry from small boats, e.g.:

1. Box, J. E., & Ski, K. (2007). *Remote sounding of Greenland supraglacial melt lakes: implications for subglacial hydraulics.* Journal of Glaciology, 53(181), 257–265. https://doi.org/10.3189/172756507782202883. Cambridge University Press & Assessment
2. Tedesco, M., & Steiner, N. (2011). *In-situ multispectral and bathymetric measurements over a supraglacial lake in western Greenland using a remotely controlled watercraft.* The Cryosphere, 5, 445–452. https://doi.org/10.5194/tc-5-445-2011. Copernicus TC

Both papers include in-situ depth sounding (Box & Ski with a raft + depth sounder; Tedesco & Steiner with a remotely controlled boat equipped with GPS and sonar).

32. You say 'in polar regions' here. But I'd suggest focussing on the GrIS earlier in your introduction and stop referencing Arctic / Polar regions after that.

41. These references are not on the GrIS so make that clear in the sentence. You could reference other papers relevant for the GrIS and AIS.

42-3. "…integrated multispectral technology with ICESat-2 to conduct bathymetric detection and inversion, leveraging both active and passive remote sensing…" Here and elsewhere in the paper (e.g. 47-8, 64) it would be useful to stick to one order and not switch the order you mention these two types of satellite data. So here you could write "integrated multispectral technology with ICESat-2 to conduct bathymetric detection and inversion, leveraging both passive and active remote sensing". It's a trivial point but it makes for a clearer, more logical read.

44. You refer to 'the mainland' but you need to be clearer in this section that the methods you're using were first applied in settings outside the GrIS. Tell us what mainland here as it reads like you're talking about mainland Greenalnd!

46. "…leverages active and passive remote sensing techniques…" You told us that 3 lines up so delete. More useful to tell us what sensors .

50. Full stop after 'Bermuda."

50. 'proposed' is the wrong word. Do you mean 'divided'? Or 'classified'?

54. You say "improving the inversion accuracy" but compared to what?

56-7. Before the sentence spanning these lines, I'd specify that you're focussing on surface lakes on ice masses. Note it's not just for 'Arctic regions' as you include Fricker et als work on Antarctic surface lakes in your referneces.

61. "Watta algorithm". This is the 3rd algorithm that's been introduced now with no details. It'd be more important to tell us the basis of this (and earlier) algorithms not what Datta and Wouters called it. I assume they called it this because it is an amalgamation of their names, but this is not really crucial information here.

61. YOU say 'arctic lakes' but it was just for GrIS lakes. Having already focussed down on the GrIS in your review, I'd stick to referencing work relevant to the GrIS here and not keep mentioning 'Arctic' or 'Polar regions' etc.

63. Delete 'some' and delete 'from 2019 to 2023' This latter not relevant.

65. 'polar' See my comment above for line 61.

65-7. Is this just the Lv et al (2024) paper that's relevant here? If so say so. But you should also acknowledge Mousavi et al, Pope et al, Williamson et al, Melling et al, etc. who did consider different bands using the 'Philpot' algorithm.

68 Suggest "…Chu et al. (2023) to offshore islands…"

68-70. So you're building on Lv et al 2024 which so far is the only paper to apply what you're calling the SDB method to GrIS lakes, but you're adding spectral stratification as used by Chu et al in a different context? Could state this more clearly

72-3. Ok so you're 'calibrating' using ICESat-2 and validating with ArcticDEM. This could be expressed more clearly.

74 'Arctic' => 'GriS'

74. "...offers effective technical support for predicting Arctic glacier melt and global climate change." This is too grand. I'd delete this. You can't do this based on your work.

75-7. Suggest delete.

80-2. Suggest delete "in the Arctic, the second-largest ice sheet in the world, surpassed only by the Antarctic Ice Sheet. However, the GrIS is more fragile and sensitive to temperature changes than the Antarctic Ice Sheet (Robinson et al., 2012)." Everybody knows the first statement and the 2$^{nd}$ is a bit vague.

85. 'aimed => 'aims' and "verify" => 'validate'

86 "bathymetry data ...were used" [data are plural]

91. Its blue green red yellow respectively

Fig 1. Your Arctic inset is rather ugly. I suggest use another inset - just for the GrIS - and remove the words 'study area' from the map.

100-1. Suggest change to "Sentinel-2 imagery was obtained for lakes A and B on 4 July 2020, lake C on 17 July 2022, and lake D on 15 July 2021."

102. 'can be' => 'was' [say what you did not what is possible to do]

110-11. Suggest "The left and right points of each beam pair are approximately 90 m apart in the transverse track direction and about 2.5 km apart in the along-track direction."

112-13. Suggest : "The ICESat-2 data used in this study were acquired on 6 July 2020 (lakes A and B), 15 July 2021 (lake C), and 14 July 2022 (lake D)."

115 "data, and"

115-16. "Please note that this study assumes" => "We assume"

118 "can be" => "data were"

119 delete "and has the characteristics of a large coverage area and high spatial resolution."

121. Delete "high-resolution Arctic digital elevation model ArcticDEM"

122-4. Ambiguous. Tell the reader what you did - what data you used.

124. Delete "It should be pointed out that"

124-5. Explain why these dates. Presumably the first data to cf. with your derived bathymetry along the ICESat-2 line for lakes A and B, 2nd for bathymetry of lake C, 3rd for lake D?

130-1. You say "To address the challenges and the limitations of traditional bathymetric methods, which do not consider the varying penetration of electromagnetic waves of different wavelengths into water…" But this is not true as many previous studies using the 'Philpot' method have done this.

Figure 2. This implies you're producing two versions of the Lyzenga model - traditional and spectral stratified. And you'll compare them both against Arctic DEM data? This was not mentioned as part of your methodology earlier in the abstract or in your brief overview of the methodology for the work on lines 68-73. It should be clearer earlier that this is part of your work.

140. Delete "provided by the ESA" and say "…data are…"

142. '…can be processed…' Again, tell us what you did not what can be done.

144-6. Why did you not use the version for separating water from ice? NDWI_ice? there's another variant of NDWI tailored for glacier ice detection that uses the blue and red bands instead of green and NIR.
That formulation is:
NDWIice =(Blue-Red)/(Blue+Red)
- Blue: reflectance in the blue band (~0.45 µm, e.g., Sentinel-2 Band 2)
- Red: reflectance in the red band (~0.65 µm, e.g., Sentinel-2 Band 4)

151. 'data are'

152. Delete "In this study"

154 and 155 "should say 'the four'

155 "Due to the fact that" => 'Because'

157-8. "Finally, more accurate bathymetric photons were obtained for constructing a bathymetry inversion model." I don't understand this sentence and it splits up the two either side of it that are related. I suggest delete this sentence (or clarify what it means if it's important).

169. Delete "It should be noticed that"

176-7. Suggest " This study applied the spectral stratification method using the Otsu algorithm, which automatically determines thresholds without requiring input parameters (Otsu, 1975)."

177-8. This is not a sentence. The main clause is missing a finite verb. Right now, "multispectral images of water stratified into four layers" is written as if it's a complete statement, but it lacks a clear subject performing an action. Should it say '…water were stratified…" ?

But 'stratified into 4 layers' implies lake depth layers, but you don't mean this - you're giving 4 wavebands. This is confusing.

179. Could delete 'layer'

189. It'd be useful to show the equation for this "traditional Lyzenga model" too.

191. You say "…combining the traditional Lyzenga model…" But with what? This is unclear.

192-3. "the near-infrared layer, the red layer, and the green layer were combined for processing. In other words, Arctic SGLs were divided into green and blue layers." This is contradictory. Did you use the NIR and Red or just the Green? Explain more precisely what you did.

197-8. So what values do these parameters take? You haven't really explicitly stated you're fitting these equations using the ICESat-2 data to derive Z, then you're using these calibrated equations to derive bathymetry for the whole lake. Please in your methods explain the calibration process explicitly. In the results it would be useful to know what values these parameters take for the two equations for the 4 lakes. It'd also be useful to know what the equation for the original Lyzenga model is, how that works, and what the parameter values are for that and how they vary between lakes.

210 'performed' => 'performs'

211 'was' => 'is'

213. It's only here that it becomes apparent that you're constructing 2 models. The original Lyzenga and a spectral stratified version of Lyzenga? This has not been clear throughout your methodology sections so far. For example, you do not mention this in Section 3.

216-22. This should all have been stated above in methods. Not here. Start the results section with the results!

229. Where have you used the active (ICESat-2) data to derive Fig 5? This was not adequately explained.

231. You say "…underscoring the reliability and feasibility of the spectral stratified model". But you cannot conclude this until you've compared with the Arctic DEM. Delete this.

232. You don't need any of these phrases ending with the word 'that', e.g. 'It should be noted that…'. Just delete this and all such phrases.

232-3. You say " the ArcticDEM data only contains spatial information of the lake bottom, and lacks water surface elevation information when obtaining bathymetry benchmark data". Is that true? Esp as you use early season strips. How do you know the lakes were empty at that time rather than containing water and frozen over?

236-7. You say "However, the sediment in the lakes of the experimental area primarily consists of bedrock, a type of material that remains stable and does not undergo significant changes over short periods." This makes no sense to me at all.

239. Having read to here, I'm not convinced you needed any of the previous section 4.1 on the qualitative analysis. Consider deleting it as the quantitative analysis is what is needed.

242. "spectral stratified Lyzenga model" . Check entire paper and refer to this model in the same way throughout. This is quite clear here but you've not so far ever referred to it in this way.

246. 'visually demonstrate' => "illustrate"

247-9. This sentence is obvious and could be deleted.

249-50. Do you mean to refer to Fig 7 here as it doesn't show ArcticDEM validation. Do you mean Fig 8? If so refer to Fig 7 earlier.

251. " spectral stratified model" See my comment for line 242. Try to refer to this model consistently throughout the paper.

256-7. You say "...are primarily concentrated in the 2-6 m range, with more pronounced differences at the transitions between different spectral layers. This is hard to see, esp. the last point as you don't mark on where the different spectral layer data were used to construct the bathymetry.

Fig 8. I'd assumed this would be a subset of Fig 6a, but for the data in the 2 sub-areas shown in Fig 7. But it's 'density' . Why are you showing this? It needs explaining.

276-7. Delete this sentence from here. This belongs in the Abstract and the start of the Conclusion.

278. You say "volume changes" but this implies you're going to determine volume change through time, i.e. applied your final model over more time periods. But you don't do this, which is a shame. Clarify what you mean by 'volume changes' here.

Figure 9. I'm not fully convinced these add anything although they're nice visually. Suggest put in Supplementary Materials. But what would be more useful to know is what are the volumes of the 4 lakes derived by the traditional Lyzenga method and from the Arctic DEM? How do the 3 estimates compare? Which overestimates vs. underestimates cf. others?

289-90. You say "While the spectral stratification-based method enhances the accuracy of bathymetric inversion compared to traditional approaches, such as the Lyzenga model..." But you can't say this. You can't generalise. You can only refer to the traditional Lyzenga model here, not ALL 'traditional approaches' by which I assume you also mean the Stumpf model? As that is the only other one you mentioned in your Intro. Did you ever consider applying this one and comparing it? And of course it'd have been valuable and interesting to have applied the 'Philpot' method too and evaluated that.

291. 'amount' => 'number'

291. What do you mean 'strip-shaped data'? Do you mean " orbital track" ?

291-2. You say there were not enough training sample data. But each lake seems well sampled along the tracks according to Fig 3. What was the total number of sampling points for each lake? Are you saying you need more? Is that really the case?

292. You say the lack of training data "...hindered further improvement in model accuracy" which is poorly phrased but more importantly, you can't conclude that. Perhaps extra data would not have improved accuracy (when cf. ArcticDEM). It MAY have improved the estimation of the model parameters and thereby led to greater accuracy, but you don't know this.

293-4. You say "...where bathymetric variations are minimal, the impact of spectral penetration differences is limited, resulting in only marginal improvements in accuracy." Did you point out such areas? Looking at Fig 6 I do not see greater 'improvements' at high depths cf. shallow depths.

294. 'Second,...'

---

## Author Comment (AC1)

Dear community reviewer:

Thank you very much for reviewing this manuscript. We have revised and corrected the manuscript according to your suggestions, and have responded to each of your questions in detail. In this response, your questions and suggestions are marked in black, while my responses and the revised sections of the manuscript based on your suggestions are marked in blue.

Traditional bathymetric methods, such as airborne LiDAR and shipborne sonar, face significant difficulties and limitations when acquiring bathymetry data for Arctic supraglacial lakes. Using remote sensing as a non-contact method to obtain bathymetric information of Arctic supraglacial lakes is emerging as a promising approach and is gaining increasing attention. This paper combines ICESat-2 data and Sentinel-2 multispectral imagery, and applies a spectral stratification strategy to derive the bathymetry, achieving a high accuracy water depth of four representative lakes. Generally, this paper is well-written and has an easy-to-follow structure. However, there are still several points in the paper that need attention.

Major issues:

**R1Q1:** The paper proposes a spectral stratification method combined with the Lyzenga model, but the mechanism and rationale for combining near-infrared, red, and green bands into two layers (green and blue) are not fully explained. It is recommended to add a brief explanation to enhance the clarity of the methodology.

**Major issues R1A1**: Thank you very much for the comment, which makes this manuscript more rigorous. In this manuscript, the red band, near-infrared band, and green band were indeed merged, and the remote sensing images were divided into only two parts: the merged layers and the blue

band layer. This part of the manuscript lacked sufficient description, and the revised paragraph is as follows:

30      As shown in Figure 1, the segmentation results for Lake A indicate that the segmented areas in the near-infrared and red bands are significantly smaller compared to those in the blue and green bands. This small region results in fewer ICESat-2 data points intersecting the NIR and red band segments, thereby reducing the accuracy of bathymetric model construction. To mitigate this limitation, the study employs a band-merging approach, combining the NIR, red, and green bands

35   for analysis. While this approach may appear to overlook the differences in penetration capability among the red, near-infrared, and green bands, it actually enhances the continuity of model training by incorporating a larger amount of ICESat-2 data. Comparative analysis shows that this integrated approach yields more stable bathymetric inversion results than processing the NIR and red bands separately.  In other words, Arctic supraglacial lakes (SGLs) were divided into layer of the NIR,

40   red, and green bands and layer of blue band for bathymetry inversion, the expression was (Lyzenga 1978, 1985; Chu et al., 2023):

$$Z = a_{g0} + \sum_{i=1}^{N} a_{gi} \ln\left[L(\lambda_i) - L_\infty(\lambda_i)\right] \tag{1}$$

$$Z = a_{b0} + \sum_{i=1}^{N} a_{bi} \ln\left[L(\lambda_i) - L_\infty(\lambda_i)\right] \tag{2}$$

where $Z$ was the predicted bathymetry result, $a_{g0}$ and $a_{gi}$ were the corresponding parameters of the

45   combined segmented image layer of the NIR, red, and green bands, $a_{b0}$ and $a_{bi}$ were the corresponding parameters of the blue band layer, $L(\lambda_i)$ was the reflectance value of the $i\text{-}th$ band, and $L(\lambda_\infty)$ was the reflectance of the deep-water zone in the $i\text{-}th$ band.

[Figure]

**Figure 1.** Binarized segmentation and spectral stratification of the Lake A

50

References

[1]      Chu, S., Cheng, L., Cheng, J., Zhang, X., and Liu, J.: Shallow water bathymetry using remote sensing based on spectral stratification, Haiyang Xuebao, 45, 125-137, http://doi.org/0253-4193(2023)01-0125-13, 2023.

[2]      Lyzenga, D. R.: Passive remote sensing techniques for mapping water depth and bottom features, Applied optics, 17, 379-383, http://doi.org/10.1364/ao.17.000379, 1978.

[3]      Lyzenga, D. R.: Shallow-water bathymetry using combined lidar and passive multispectral scanner data, International journal of remote sensing, 6, 115-125, http://doi.org/10.1080/01431168508948428, 1985.

55

**R1Q2**: There is a time difference of two to four months between Sentinel-2 images and ArcticDEM validation data. Although the paper mentions that lakebed materials are relatively stable, it is suggested to strengthen the explanation and explicitly acknowledge the potential uncertainty this brings.

60

**Major issues R1A2**: We sincerely appreciate the reviewer's careful reading and valuable comment. Although the lakebed in the study area is mainly composed of bedrock and is unlikely to undergo significant changes over a short period, minor variations are inevitable. Therefore, deviations between ICESat-2-derived bathymetry and ArcticDEM are indeed unavoidable. However, the overall consistency between the two datasets is strong, as also demonstrated in the experiment by Melling et al. (2024). The discrepancies fall entirely within an acceptable range and do not undermine the validity of ArcticDEM as a high-quality validation dataset. Due to the limited temporal resolution of ArcticDEM, it is difficult to conduct comparative experiments to assess the impact of different acquisition dates on the validation results. We have clarified this limitation in the revised manuscript, which is provided in the following paragraph. We also aim to address it through experimental improvements in future work.

Additionally, there was a temporal discrepancy of two to four months between the ArcticDEM data and the ICESat-2 data in this study. This time difference inevitably introduces some error between the ArcticDEM data and the ICESat-2 data used for modeling. However, the sediment in the lakes of the experimental area primarily consists of bedrock, a type of material that remains stable and does not undergo significant changes over a short period, Melling et al. (2024) also demonstrated through their experiments that ICESat-2 and ArcticDEM data collected several months apart show a high degree of consistency. Therefore, under appropriate conditions, it is feasible to use ArcticDEM as validation data in this study (Melling et al., 2024)

**References**

[1]     Melling, L., Leeson, A., McMillan, M., Maddalena, J., Bowling, J., Glen, E., Sandberg Sørensen, L., Winstrup, M., and Lørup Arildsen, R.: Evaluation of satellite methods for estimating supraglacial lake depth in southwest Greenland, The Cryosphere, 18, 543-558, http://doi.org/10.5194/tc-18-543-2024, 2024.

**R1Q3**: The implementation details of NDWI are missing: the Methods section mentions using NDWI for water-land separation but does not specify the exact Sentinel-2 band numbers (e.g., B3 for Green and B8 for NIR), which may affect the reproducibility.

**Major issues R1A3:** Thank you very much for the suggestion, which makes this manuscript clearer. Indeed, the manuscript does not specify the correspondence between different wavelength ranges and the Sentinel-2 band numbers. Generally, the green band corresponds to Band 3 of Sentinel-2, and the near-infrared band corresponds to Band 8. We have clarified in the revised manuscript as below:

The water column was extracted from multispectral imagery using water-land separation methods, i.e., the Normalized Difference Water Index (NDWI) (Eq. (3)) and threshold-based grayscale segmentation (McFeeters, 1996)

$$NDWI = \frac{Green - NIR}{Green + NIR} \tag{3}$$

where *Green* represents the reflectance at the green band (corresponding to Band 3 of Sentinel-2), and *NIR* represents the reflectance at the near-infrared band (corresponding to Band 8 of Sentinel-2). Since the multispectral imagery obtained contained partially unmelted ice cover on the acquisition date, a mask was applied to the ice cover on the lake to ensure the accuracy of the study during the water column extraction process.

**References**

[1]     McFeeters, S. K.: The use of the Normalized Difference Water Index (NDWI) in the delineation of open water features, International journal of remote sensing, 17, 1425-1432, http://doi.org/10.1080/01431169608948714, 1996.

115 **R1Q4**: The validation focuses on RMSE and MAE, but lacks a more detailed visualization of residual distribution (e.g., error scatter or residual histograms). It is suggested to add one simple figure or a few lines of text to further illustrate the validation quality.

**Major issues R1A4:** Thank you very much for your insightful comments, which make this
120 manuscript more rigorous. We have revised the manuscript to include a more comprehensive explanation in this part, the revised paragraph is as follows:

To quantitatively and visually demonstrate the performance of the method, the validation results of the SDB using the traditional Lyzenga model and the spectrally stratified Lyzenga model
125 are presented in Figure 2.

[Figure]

(a)

[Figure]

(b)

(c)

(d)

[Figure]

**Figure 2.** Comparison of bathymetric inversion results using the Lyzenga model and spectral stratified Lyzenga model for four lakes. (a) Lake A bathymetry validation. (b) Lake B bathymetry validation. (c) Lake C bathymetry validation. (d) Lake D bathymetry validation. The point represents the ArcticDEM validation points, the black dashed line indicates the 1:1 line, and the red solid line represents the data fitting line.

130      To visually demonstrate the improvement in the bathymetric inversion accuracy achieved by the spectral stratification method, we selected Lake A as a case study to compare the traditional Lyzenga model with the spectral stratified Lyzenga model. Based on the bathymetric inversion results shown in Figure 3, the study examines the differences in depth inversion results between the two methods. In areas where relatively obvious discrepancies are observed, ArcticDEM data

135    is used for accuracy validation, as illustrated in Figure 4.

**Figure 3.** Comparison of the bathymetric maps using the Lyzenga model and the spectral stratified model with Lake A. (a) The depth inversion results derived from the Lyzenga model. (b) The depth inversion results are derived from the spectral stratified model. The red box highlights areas with relatively notable differences in depth inversion between the two models. (c) A detailed view of the water area within the red box.

140    A qualitative analysis of the bathymetric inversion results for Lake A indicates that the discrepancies between the Lyzenga model and the spectral stratified model are primarily concentrated in the 2-6 m range, with more pronounced differences at the transitions between different spectral layers.

[Figure]

(a)                                                 (b)

Density *Low* ▓▓▓▓▓▓▓▓▓▓▓ *High*

**Figure 4.** Comparison of accuracy using the ArcticDEM data with the Lyzenga model and the spectral stratified
145  Lyzenga model of Lake A. (a) The Lyzenga model. (b) The spectral stratified model. The black dashed line indicates the 1:1 line, and the red solid line represents the data fitting line.

According to Figure 2, $R^2$ is relatively high, all of which could reach 0.9 or above, proving that the Lyzenga model is rational for application in these Arctic SGLs. Specifically, the RMSE of the bathymetry inversion for Lake A decreased from 0.54 m to 0.47 m, and the MAE decreased
150  from 0.43 m to 0.37 m, with reductions of 13.0% and 14.0%, respectively. For Lake B, the RMSE and MAE of the spectral stratified bathymetry inversion accuracy values were 0.61 m and 0.46 m, respectively, with decreases of 9.0% and 13.2%. In the case of Lake C, the RMSE was reduced to 0.50 m, with reductions of 7.4%. For Lake D, the RMSE reduced to 0.72 m and MAE reduced to 0.52 m, which the RMSE decreased by 5.3% and MAE decreased by 8.8%. Validation results for

the red box region of Lake A indicate that the spectral stratified method achieves higher accuracy than the traditional Lyzenga model too, with the RMSE reduced from 0.67 m to 0.47 m, a decrease of 29.9%, and the MAE reduced from 0.59 m to 0.39 m, a decrease of 33.9%, as shown in Figure 4. The validation results shown in Figures 2 and 4 indicate that the spectrally stratified bathymetry depth inversion model achieves higher accuracy across different depth ranges by taking into account the varying penetration capabilities of electromagnetic waves at different wavelengths. Taking Lake A as an example, and based on validation with ArcticDEM, the scatter plots clearly demonstrate that the spectrally stratified model outperforms the traditional Lyzenga model in terms of accuracy across various water depths. For the points with significant absolute errors in the validation results, this may be related to anomalies in pixel reflectance values caused by errors during the radiometric correction of the multispectral imagery data. The validation results further demonstrate the effectiveness of the spectral stratification method in improving bathymetric inversion accuracy. In summary, the experimental results demonstrate that the proposed spectral stratified model can comparatively improve the bathymetric accuracy for Arctic SGLs by the SDB method.

**R1Q5**: In the Discussion section, although the challenges and limitations are mentioned, the dynamic changes of supraglacial lakes are described relatively generally. It is suggested to slightly expand the discussion on how data acquisition timing affects inversion accuracy.

**Major issues R1A5:** We appreciate your suggestion, as it is a crucial point in the manuscript. The morphological changes of supraglacial lakes in Arctic regions occur rapidly, and training datasets from different dates can significantly affect the SDB results. This differs from the SDB methods that are widely used for ocean bathymetry. According to your valuable comment, we have modified the Discussion section, as follows:

The dynamic nature of Arctic SGLs, which undergo significant morphological changes over short periods, as shown in Figure 5, where the lake's morphology changed markedly within just 7 days, this poses a challenge for combining ICESat-2 and Sentinel-2 data in bathymetric inversion. Accurate results require close temporal synchronization of these datasets, which imposes higher demands on data acquisition and availability.

[Figure]

**Figure 5.** Comparison of the morphology of Lake A on different dates: (a) Sentinel-2 image on June 27, 2020; (b) Sentinel-2 image on July 4, 2020.

Minor issues:

**R1Q6**: Minor grammatical and typographical errors exist in the manuscript. For example, "Figure. 1" should be "Figure 1" without a period

**Minor issues R1A6:** Thank you very much for this comment, which makes this manuscript more rigorous. We have revised the entire manuscript to correct similar instances.

195 **R1Q7**: In Section 3.1.2, the font of the section number "3.1.2" should be standardized to Times New Roman.

**Minor issues R1A7:** Thank you very much for this comment, which makes this manuscript more rigorous. We have thoroughly checked the entire manuscript to ensure that no such detail issues 200 remain.

**R1Q8**: In line 283, results such as 9.46 $10^6$m$^3$ should have a space between the number and the unit.

205 **Minor issues R1A8:** Thank you for your comment, which enhances the rigor of this manuscript. We have thoroughly revised the entire manuscript to address similar instances.

**R1Q9**: The date format in Table 1 should be clearly indicated as dd/mm/yyyy to avoid ambiguity.

210 **Minor issues R1A9:** Thank you very much for your suggestion, which makes this manuscript clearer. We have made corrections to the manuscript to ensure rigor and clarity before its publication.

**R1Q10**: The terms "spectral stratified model" and "spectral stratification model" in the text should 215 be unified into a single expression.

**Minor issues R1A10:** Thank you very much for this comment, which makes this manuscript more rigorous. We have unified the usage of the "spectral stratified model" throughout the manuscript.

220 In this manuscript, the sections that need improvement have been highlighted in blue. Your comments and have greatly contributed to enhancing the quality of this manuscript. Thank you once again for taking the time to review it.

Best regards.

---

## Author Comment (AC2)

**Responses to comments from Referee 1:**

Dear Referee,

Thank you very much for reviewing this manuscript. Your professional insights have been constructive for improving the manuscript. We have basically revised and corrected the manuscript according to your suggestions and have responded to each of your comments in detail, including revisions to the textual descriptions, redesign of relevant sections, and updates to the experimental content. In the document, your comments and suggestions are marked in black, while our responses and the revised sections of the paper based on your suggestions are marked in blue.

**General comments**

The paper presents a novel approach for estimating the bathymetry of supraglacial lakes by integrating ICESat-2 laser altimetry with Sentinel-2 multispectral imagery. The method combines spectral stratification ("Otsu algorithm") and a classical regression model ("Lyzenga model") and is tested on four lakes in Southwest Greenland. The topic is timely and relevant, and the authors' effort to improve bathymetric inversion accuracy is commendable. The results show modest improvements in accuracy compared to the Lyzenga model without spectral stratification, which are encouraging.

However, the manuscript requires substantial revision before it can be considered for publication. In particular, the methodology needs to be presented more clearly and systematically, with sufficient detail to allow reproducibility and critical evaluation. Below are the main concerns that should be addressed.

We sincerely appreciate your thorough and professional review of the manuscript and the valuable comments you have provided. Your recognition of the research content is highly encouraging for us. We have carefully considered all the comments and questions you raised, and in the following responses, we provide detailed explanations and replies to each of them.

**Q1: Title and scope:**

The current title implies broad applicability across Arctic supraglacial lakes, which is not supported by the limited dataset used in the study. Since the method is only applied to four lakes in Southwest Greenland, the title should be revised to reflect this scope more accurately. Additionally, the discussion should include a thorough assessment of the method's generalizability to supraglacial lakes on other regions of the Greenland Ice Sheet (and other Arctic ice masses), including potential limitations.

**A1:** Thank you so much for your valuable comments. The study area selected in this work consists of four supraglacial lakes in southwest Greenland, which is not sufficient to represent the entire Greenland Ice Sheet or the broader Arctic region. Therefore, we have revised the title to 'Southwest Greenland supraglacial lake bathymetry derived from ICESat-2 and spectral stratification of satellite imagery' to achieve better alignment between the title and the content. In addition, we have supplemented the discussion section to further explore the applicability of the proposed method over larger spatial scales.

**Q2: Methodological clarity:**

Several key components of the methodology are insufficiently described:

Lyzenga Model: This model is central to the study but is only briefly introduced. A more comprehensive explanation is
needed, including its assumptions (e.g., uniform water clarity, bed type, and spectral behavior). Clarifying these
assumptions is essential for understanding the model's applicability and limitations.

**A2 (1):** Thank you so much for your valuable comments. As you pointed out, the Lyzenga model plays an important role in this study, and it is therefore necessary to provide further clarification of the model. The Lyzenga model is derived from the Beer–Lambert law, in which certain parameters are empirically determined, resulting in its final mathematical expression. This

model is particularly suitable for bathymetric inversion in relatively shallow waters. By incorporating partial in situ measurements as constraints, the model parameters can be estimated using algorithms such as the Levenberg–Marquardt (LM) method. The Lyzenga model has been successfully applied in clear oceanic waters and inland lakes. Please see the revised contents as follows:

**3.2.2 Traditional Lyzenga model**

Both Lyzenga multi-band logarithmic linear model (Lyzenga, 1978, 1985) and Philpot RTE models exploit the exponential attenuation of light in water following Beer-Lambert's law, with Philpot's RTE approach providing explicit physical parameterization of bottom reflectance and water column properties that were empirically combined in Lyzenga's earlier formulation. Unlike the purely theoretical RTE approach, which derives water depth only from optical imagery, this method introduces empirical constraints using a limited number of measured depth values to estimate model parameters. In this formulation, the bottom reflectance ( $A_d$ ) and the diffuse attenuation coefficient function (g) are treated as constants, where  $a_0 = ln(A_d - R_\infty)/g$ ,  $a_1 = -1/g$ . At this stage, the model equation is written as:

$$Z = a_0 + a_1 \left( R_{\omega} - R_{\infty} \right) \tag{1}$$

By integrating the spectral reflectance information from multiple bands, the following expression can be obtained:

$$Z = a_0 + a_i \sum_{i=1}^n \ln \left[ R_{\omega}(\lambda_i) - R_{\infty}(\lambda_i) \right]$$
 (2)

This represents the commonly used Lyzenga model. In this study, ICESat-2 bathymetric points are incorporated to constrain the empirical parameters of the Lyzenga model, and the parameter estimation is performed using the Levenberg–Marquardt (LM) algorithm.

**Reference:**

- [1] Lyzenga, David R. "Passive remote sensing techniques for mapping water depth and bottom features." Applied optics 17.3 (1978): 379-383.
- [2] Lyzenga, David R. "Shallow-water bathymetry using combined lidar and passive multispectral scanner data." International journal of remote sensing 6.1 (1985): 115-125.
- Use of ICESat-2 data: The role of ICESat-2 data in the workflow is unclear. The abstract does not mention it, and the main text provides only a brief reference to its use as training data. It is important to specify how the data is used for calibration of the Lyzenga model: Which parameters are calibrated? And is data from both weak and strong ICESat-2 beams considered. Furthermore, it should be clarified whether the photons identified as originating from the bed are used directly as discrete point data, or if this point cloud is processed into a continuous bathymetric surface beforehand (such as through fitting a smooth model).

**A2 (2):** Thank you so much for your valuable comments, which have helped make the manuscript more rigorous. It is necessary to clarify several points here. In this study, ICESat-2 data were used to calibrate the empirical parameters of the Lyzenga model, with both strong and weak beam tracks utilized. We have supplemented the manuscript with complete ICESat-2 information, including track details, as shown in Table 1 (the corresponding Table 2 in this document is provided in A34). Moreover, ICESat-2 was applied as discrete point data rather than being fitted into a continuous curve. These clarifications have been fully incorporated into the revised manuscript in the sections on ICESat-2 data processing and model construction.

Otsu Algorithm: The algorithm is mentioned without adequate explanation. While some details are provided in Section 3.2,
 a clearer and more complete description of how the algorithm is applied to spectral stratification would benefit readers unfamiliar with this technique.

**A2** (3): Thank you so much for your valuable comments, which have helped make the manuscript more rigorous. We have added further explanation of the Otsu algorithm in the manuscript to make its role in spectral stratification easier to understand. Please see the revised contents as follows:

This study applied a spectral stratification method based on the Otsu algorithm, which automatically determines the optimal threshold without requiring user-defined parameters. The algorithm adaptively selects the threshold according to the statistical distribution of pixel intensities in the image histogram by maximizing the between-class variance and minimizing the within-class variance (Otsu, 1975). By utilizing the penetration characteristics and reflectance differences of water across various spectral bands, multispectral images of water were stratified into four layers: near-infrared band, red band, blue band, and green band.

**Reference:**

[3] Otsu, Nobuyuki. "A threshold selection method from gray-level histograms." Automatica 11.285-296 (1975): 23-27.

• Training strategy: The manuscript does not specify whether the model is trained individually for each lake or whether data from all lakes are pooled to form a single training dataset. Clarifying this point is essential for evaluating the robustness and scalability of the approach. The authors note (L291) that the method is constrained by the limited availability of ICESat-2 data for training. However, if ICESat-2 data from, say, hundreds of lakes are pooled, the volume of training data could be significantly increased. Conversely, if the model is trained separately for each lake, its applicability would be restricted to lakes directly intersected by an ICESat-2 track, limiting its broader utility.

**A2 (4):** Thank you so much for your valuable comments; your suggestion provided us with a fresh perspective. The model used in this study is trained separately for each lake. Admittedly, as you noted, if a large number of ICESat-2 datasets (e.g., hundreds of tracks) were used for training, the model's stability would undoubtedly improve. In this study, we chose to build models individually for each lake because the spectral characteristics of the water bodies (i.e., the relationship between water depth and reflectance) vary across different lakes. By dividing the optical imagery into zones based on spectral penetration differences and constraining model parameters with a subset of in situ bathymetry data, we aim to achieve higher inversion accuracy for each lake. Moreover, this spectral-stratified zonal inversion approach is transferable and expandable, and expanding the model's transferability and applicability will be an important focus of our future work.

**Q3: Comparison with other methods:**

The manuscript would benefit from a broader comparison with recent approaches, such as those proposed by Datta et al (2021) and Melling et al. (2024). Including validation metrics from these methods – applied to the same four lakes – would help contextualize the contribution of the present study and highlight its strengths and limitations relatively to existing techniques. At present, the two models compared in the manuscript – the classical Lyzenga model and the spectral stratified variant – are closely related and yield relatively similar performance. A comparison with more distinct methodologies would provide a more meaningful benchmark and significantly enhance the scientific value of the study.

**A3:** Thank you so much for your valuable comments. We supplemented the analysis with the radiative transfer equation (RTE) model proposed by Philpot (1987) to perform bathymetric inversion and accuracy validation for the four lakes, following the approach adopted by Lutz et al. (2024). Corresponding contents have been made in the manuscript as follows:

**3.2.1 Radiative transfer equation model**

The RTE model for bathymetry inversion was proposed by Philpot (1987). It is derived based on Beer-Lambert's law and has been widely applied in oceans and lakes. Its expression is shown in Equation (3) (originally Equation (2) in the manuscript):

$$Z = g^{-1} \left[ \ln \left( A_d - R_{\infty} \right) - \ln \left( R_{\omega} - R_{\infty} \right) \right]$$
(3)

Where  $R_{\omega}$  is the water surface reflectance;  $A_d$  is the bottom reflectance; g is a function of the diffuse attenuation coefficients for upward and downward radiance;  $R_{\infty}$  is the reflectance of infinitely deep water; and Z is the water depth.

In Philpot's RTE equation, the parameter determination follows previous studies. Specifically, Ad is calculated from averaging the reflectance values within a 30 m radius around each lake (Moussavi et al., 2020). However, ideal optically deep waters are typically absent within the GrIS region, making it difficult to obtain the reflectance values of spectral bands at infinite water depth. Therefore, following the approach of Melling et al. (2024), this study references the deep-water reflectance values of corresponding bands derived from multiple other Sentinel-2 scenes as substitutes. The values of g are typically adjusted to match the specific wavelengths observed by different satellite missions. The RTE equation is constructed using the reflectance of the green band due to its extended depth range and consistent depth-reflectance relationship up to approximately 10 m, following the approach of Lutz et al. (2024). Accordingly, the corresponding g value for the green band is 0.1413, as determined by Williamson et al. (2018) for Sentinel-2.

**References:**

- [4] Philpot, William D. "Radiative transfer in stratified waters: a single-scattering approximation for irradiance." Applied Optics 26.19 (1987): 4123-4132.
- [5] Moussavi, Mahsa, et al. "Antarctic supraglacial lake detection using Landsat 8 and Sentinel-2 imagery: towards continental generation of lake volumes." Remote Sensing 12.1 (2020): 134.
- [6] Melling, Laura, et al. "Evaluation of satellite methods for estimating supraglacial lake depth in southwest Greenland." The Cryosphere 18.2 (2024): 543-558.
- [7] Lutz, Katrina, et al. "Assessing supraglacial lake depth using ICESat-2, Sentinel-2, TanDEM-X, and in situ sonar measurements over Northeast and Southwest Greenland." The Cryosphere 18.11 (2024): 5431-5449.
- [8] Williamson, Andrew G., et al. "Dual-satellite (Sentinel-2 and Landsat 8) remote sensing of supraglacial lakes in Greenland." The Cryosphere 12.9 (2018): 3045-3065.

**Q4: Evaluation and visualization:**

The evaluation of the method relies solely on scatter plots comparing estimated water depths with ArcticDEM-derived values. While informative, this approach would benefit from being complemented by spatial difference maps to visualize where discrepancies occur. Such maps could reveal whether both models struggle in the same regions and under what conditions, offering insights into potential sources of error and guiding future improvements.

**A4:** Thank you so much for your valuable comments, which have helped make the manuscript more rigorous. Relying solely on scatter plots of error distribution to evaluate the accuracy of the two bathymetric inversion models may not be sufficient. Therefore, this manuscript has been supplemented with spatial maps of bathymetric errors for the four lakes under study — not only for the two models in the original manuscript, but also for the newly added Philpot RTE model — along with a more in-depth analysis of the sources of error, as shown in Figure 1 (corresponding to Figure 6 in the original manuscript)

Figure 1. Differences between the water results derived from the three SDB models and those obtained from ArcticDEM. Red areas indicate overestimation, while blue areas indicate underestimation.

**Summary**

In summary, the manuscript addresses an important topic and presents a promising approach. However, substantial revisions are needed to improve clarity, methodological transparency, and contextualization. I encourage the authors to restructure the presentation of the method, provide more detailed descriptions of key components, and expand the discussion to include broader applicability and comparative analysis.

Your comment is constructive for improving the manuscript and makes it more scientifically sound and rigorous. Based on your suggestion, combined with our own understanding, we will comprehensively revise the descriptions of methods, discussions, and related sections.

**Specific comments**

**Q5:** L16 + L84: The manuscript states that the four lakes are "representative," but it is unclear what they are representative of. Are they meant to reflect characteristics of all supraglacial lakes across the Greenland Ice Sheet? Please clarify the criteria used for selecting these lakes and substantiate the claim of representativeness.

**A5:** Thank you so much for your valuable comments, which have helped make the manuscript more rigorous. The selected lakes are four relatively large and morphologically intact lakes on the southeast GrIS. These lakes provide favorable conditions for acquiring subaqueous bathymetric points from ICESat-2 and are suitable for applying the method presented in this manuscript. Therefore, they can be considered representative. The corresponding parts in the manuscript have been updated accordingly as follows:

To validate the effectiveness of the proposed method, we apply it to four large and morphologically intact lakes on the southeast Greenland Ice Sheet (GrIS), using time-stamped ArcticDEM (Arctic Digital Elevation Model) strips as reference data.

**Q6:** L13: In the abstract, the Otsu algorithm is referred to as the "maximum between-class variance method," but the paper does not explain what this entails or how the algorithm functions. A clear description should be added to the methods section.

**A6:** Thank you so much for your valuable comments, which have helped make the manuscript more rigorous. We have added a more detailed description of the Otsu method in the main text to help readers better understand it. The corresponding revisions in the manuscript are presented earlier in A2 (3).

**Q7:** L22: The introduction would benefit from a more detailed explanation of the role of supraglacial lakes in glacier dynamics, particularly their potential to rapidly route meltwater to the glacier bed during drainage events.

**A7:** Thank you so much for your valuable comments, which have helped make the manuscript more rigorous. We have added relevant descriptions of glacier dynamics in the introduction. The revised paragraph is as follows:

These changes in water volume within Arctic SGLs are closely linked to ice sheet dynamics through a critical mechanism: when lake depth and volume reach sufficient thresholds, rapid drainage events can deliver massive amounts of meltwater to the glacier bed, triggering basal lubrication and ice acceleration (Das et al., 2008; Stevens et al., 2015). Observations demonstrate this volume-dependent control on glacier dynamics: a drainage event at Store Glacier transferred 4.8 million m³ of water to the bed within 5 hours, accelerating ice flow from 2.0 to 5.3 m/day (Chudley et al., 2019), while cascading drainage across lake networks has produced 50-100% velocity increases over distances exceeding 80 km (Christoffersen et al., 2018). Therefore, accurate quantification of lake depth and volume is essential for understanding ice sheet response to climate warming and predicting future sea level contributions.

**References:**

- [9] Das, Sarah B., et al. "Fracture propagation to the base of the Greenland Ice Sheet during supraglacial lake drainage." Science 320.5877 (2008): 778-781.
- [10] Stevens, Laura A., et al. "Greenland supraglacial lake drainages triggered by hydrologically induced basal slip." Nature 522.7554 (2015): 73-76.
- [11] Chudley, Thomas R., et al. "Supraglacial lake drainage at a fast-flowing Greenlandic outlet glacier." Proceedings of the National Academy of Sciences 116.51 (2019): 25468-25477.
- [12] Christoffersen, Poul, et al. "Cascading lake drainage on the Greenland Ice Sheet triggered by tensile shock and fracture." Nature Communications 9.1 (2018): 1064.

**Q8:** L62: Please clarify how this work differs from the authors' previous study (Lv et al., 2024), which also uses Sentinel-2 and ICESat-2 data to derive bathymetry, seemingly for the same lakes.

**A8:** Thank you for the thorough review of our study. Our previous work applied the combined active—passive satellite remote sensing approach for bathymetric inversion to supraglacial lakes in Greenland. In this manuscript, we propose an optimization strategy for this method by introducing the concept of spectral stratification and zonal inversion, with a particular emphasis on improving bathymetric inversion accuracy.

**Q9:** Figure 1: Lakes C and D appear to be intersected by both yellow and red ICESat-2 tracks. Why is data from only one track used? Please clarify.

**A9:** Thank you so much for your valuable comments. The guiding principle for model construction in this study is to ensure, as far as possible, that the acquisition dates of Sentinel-2 and ICESat-2 data are consistent, to minimize the impact of rapid

morphological changes in the lakes. Therefore, the Sentinel-2 and ICESat-2 datasets used in this study are paired on a one-to-one basis.

Q10: L140–143: If the L1C dataset is not used in the analysis, it should be omitted from the text and from the workflow diagram in Figure 2. Only the radiometrically corrected L2A data should be mentioned.

**A10:** Thank you so much for your valuable comments, which have helped make the manuscript more rigorous. It is correct that this study did not use L1C data, and the relevant descriptions have been deleted in the manuscript.

Q11: Figure 2: The workflow diagram could be a valuable aid for understanding the method, but several steps are unclear. For example, what is meant by "water column extraction" (is this the lake area delineation?), and what does "SGLs water-leaving radiance" refer to? The figure and its caption should be revised to ensure the diagram is self-explanatory.

A11: Thank you so much for your valuable comments, which have helped make the manuscript more rigorous. We have improved the diagram to make it clearer. The term "water column extraction" refers to isolating the SGLs water portion. We acknowledge that "SGLs water-leaving radiance" is not a strictly accurate expression; it should be "SGLs water-leaving reflectance," which represents the portion of electromagnetic radiation that penetrates the water column and carries information about the subsurface reflectance. To avoid confusion, we have corrected it to "Water reflectance" in the manuscript. Given that Arctic supraglacial lakes are typically clear and possess strong electromagnetic penetration, the reflectance directly obtained from the imagery essentially corresponds to the water-leaving reflectance, making additional emphasis unnecessary. The revised workflow diagram is as follows:

Figure. 2. The workflow of the SDB method.

Q12: L148: Please explain how the ice mask is generated.

**A12:** Thank you so much for your valuable comments, which have helped make the manuscript more rigorous. We performed threshold segmentation by combining the global threshold of the grayscale image calculated using NDWIice (different from the NDWI used in the original manuscript; the corresponding experimental procedures have been revised) with manual empirical judgment. The image was divided into water and non-water parts, with the non-water (including ice covering the lake water) portion used as a mask to extract the water body. The revised contents are as follows:

The Sentinel-2 data are divided into L1C and L2A levels. The L1C level data product is a geometric precision correction radiographic product that has not undergone radiometric correction. L2A products are products that undergo radiation correction processing based on L1C. The water column was extracted from multispectral imagery using water-land separation methods, i.e., the Normalized Difference Water Index (*NDWI*), specifically its variant adapted for water extraction in ice–snow covered environments, Eq. (4) (originally Equation 1 in the manuscript) (McFeeters, 1996; Yang et al., 2012), combined with threshold-based grayscale segmentation.

$$NDWI_{lce} = \frac{Blue - Red}{Blue + Red} \tag{4}$$

Where *Blue* represents the reflectance at the blue band (corresponding to Sentinel-2 Band 2) and *Red* represents the reflectance at the red band (corresponding to Sentinel-2 Band 4).

The image was divided into water and non-water parts using the *NDWI*Ice, with the non-water portion—including lake ice—applied as a mask to extract the open-water body. As the multispectral imagery contained partially unmelted ice on the lake, this masking process effectively removed ice-covered areas and ensured accuracy during the water column extraction.

**References:**

- [13] McFeeters, Stuart K. "The use of the Normalized Difference Water Index (NDWI) in the delineation of open water features." International journal of remote sensing 17.7 (1996): 1425-1432.
- [14] Yang, Kang, and Laurence C. Smith. "Supraglacial streams on the Greenland Ice Sheet delineated from combined spectral—shape information in high-resolution satellite imagery." IEEE Geoscience and Remote Sensing Letters 10.4 (2012): 801-805.
- Q13: L172: The vertical adjustment of ICESat-2 photons is mentioned but not clearly described. Please elaborate on the rationale and procedure for this adjustment.

**A13:** Thank you so much for your valuable comments, which have helped make the manuscript more rigorous. We have provided a more detailed description of this part in the manuscript. In fact, this step was intended to reduce systematic errors caused by temporal inconsistencies in the dataset. The revised paragraph is as follows:

For details, in this study, the actual lake surface elevation corresponding to the remote sensing imagery was determined based on the location of the ICESat-2 profile intersecting the lake boundary extracted from Sentinel-2 data. The difference between the Sentinel-2 water surface boundary and the water surface photon elevation identified by ICESat-2 was then used to vertically adjust the ICESat-2-derived bathymetry. This adjustment is conceptually similar to the tidal correction applied in ICESat-2 bathymetry in oceanic settings.

Q14: L236: The statement that the lake bed consists of "bedrock" is confusing. Supraglacial lakes typically have ice beds, not bedrock. Moreover, since the ice surface evolves over time, some discrepancies between ArcticDEM and Sentinel-2 data collected 2–4 months apart are likely.

**A14:** Thank you for raising this important point. We acknowledge that our current manuscript does not provide a sufficiently clear explanation. We have revised the original text accordingly, as follows:

The GrIS is covered by ice, and both the ice surface and supraglacial lakes evolve, making short-term morphological changes unavoidable. However, as noted by Echelmeyer et al. (1991), many large supraglacial lakes remain fixed in space because their surface depressions are dynamically supported by irregularities in the underlying bedrock. This bedrock control limits large-scale spatial shifts within short periods, although minor variations are inevitable. Consequently, some discrepancies between ICESat-2-derived bathymetry and ArcticDEM data collected several months apart are expected. Nevertheless, the overall consistency between the two datasets remains strong, as also demonstrated in the experiment by Melling et al. (2024). These differences fall within an acceptable range and do not compromise the reliability of ArcticDEM as a high-quality reference dataset.

**References:**

[6] Melling, Laura, et al. "Evaluation of satellite methods for estimating supraglacial lake depth in southwest Greenland." The Cryosphere 18.2 (2024): 543-558.

[15] Echelmeyer, Keith, T. S. Clarke, and Will D. Harrison. "Surficial glaciology of jakobshavns isbræ, West Greenland: Part I. Surface morphology." Journal of Glaciology 37.127 (1991): 368-382.

Q15: Figures 6 & 8: What does the "density" color scale represent? Please clarify in the figure captions.

**A15:** Thank you so much for your valuable comments. The color bar represents point density, defined as the number of validation points within a given Euclidean neighborhood radius. The corresponding description in the manuscript has been revised in the corresponding figure captions.

**Q16:** Figure 7: Consider adding ICESat-2 track lines and difference maps comparing both models to ArcticDEM, as well as a direct comparison between the two models.

**A16:** Thank you so much for your valuable comments, which have helped make the manuscript more rigorous. We appreciate your suggestion. After considering the revised experimental content and the overall structure of the manuscript, we believe that the contents described in Figures 7 and 8 (originally L246–L260) are no longer necessary to retain. This is because we have supplemented spatial error maps for all lake model inversion results relative to ArcticDEM, and all results have been quantitatively evaluated for accuracy. These additions fully cover the information previously presented in those figures.

**Q17:** L255–258: The claim regarding discrepancies between models should be supported with visual evidence—such as a plot—to allow readers to assess this directly.

**A17:** Thank you for your suggestion. We no longer retain this section, as the updated figures and analyses already provide comprehensive visual and quantitative comparisons of all model inversion results with ArcticDEM, making the previous description redundant.

**Technical comments**

Q18: Figure captions: The figure texts (e.g., Figs 3, 5, 6, 9) are repetitive and should be written more concisely. Where appropriate, captions should be revised to ensure that they are self-contained and provide sufficient context for stand-alone interpretation.

**A18:** Thank you for your suggestion. We have revised the figure texts as suggested, making them more concise and clearer, and ensuring better consistency with the main text. Taking Figure 3 (originally in the manuscript) as an example, the revised figure and texts are as follows:

Figure. 3 Extraction and correction of ICESat-2 bathymetry photons for four lakes. (a) Data track for Lake A (b), Data track for Lake B (c), Data track for Lake C (d), Data track for Lake D.

Q19: L10 & L31: An important reason for the limited use of airborne LiDAR and shipborne sonar in supraglacial lake (SGL) bathymetry is the rapid temporal variability in lake depth. This should be explicitly mentioned alongside the logistical and environmental challenges.

**A19:** Thank you so much for your valuable comments, which have helped make the manuscript more rigorous. We have revised the manuscript and added clarifications to more fully and clearly explain the limitations of using airborne LiDAR and shipborne sonar for bathymetry in supraglacial lakes. The revised content is as follows:

Accurate lake depth measurements are essential for reliable volume estimation, yet traditional bathymetry methods (e.g., airborne LiDAR and shipborne sonar) face significant challenges and high costs in the harsh Arctic environment, and are also inadequate for capturing the rapid temporal variability in lake depth.

**Q20:** L11: Clarify here the nature of the data, i.e. specify that ICESat-2 provides photon-counting laser altimetry and Sentinel-2 offers multispectral optical imagery.

**A20:** Thank you so much for your valuable comments, which have helped make the manuscript more rigorous. We have revised the manuscript accordingly.

**Q21:** L26: Revise to "SGLs are formed \*in\* surface depressions" for grammatical accuracy.

**A21:** Thank you so much for your valuable comments, which have helped make the manuscript more rigorous. We have revised the manuscript accordingly.

**Q22:** L37: References should be placed immediately after the respective models are introduced to improve readability and attribution.

**A22:** Thank you so much for your valuable comments, which have helped make the manuscript more rigorous. We have revised the manuscript accordingly.

Q23: L41: "Precise measurement data" is vague. Specify the type of measurements (e.g., lake surface elevation, lake bed depth).

**A23:** Thank you so much for your valuable comments, which have helped make the manuscript more rigorous. We have revised the manuscript accordingly; the term measurement data specifically refers to the water depth data.

**Q24:** L43–L56: This section is overly broad, focusing on global shallow water bathymetry. It should be shortened and refocused on SGL-specific applications. Conversely, the discussion of prior SGL bathymetry methods could be expanded to better distinguish the proposed approach.

**A24:** Thank you so much for your valuable comments, which have helped make the manuscript more rigorous. We have reorganized this section to more clearly highlight the bathymetry methods specific to SGLs and to expand the comparison with other related approaches. Corresponding revisions have been made in the manuscript as follows:

In recent years, the launch of the spaceborne single-photon altimetry satellite ICESat-2 (Ice, Cloud, and Land Elevation Satellite-2) has partially mitigated challenges in obtaining precise water depth measurement data (Albright and Glennie, 2020; Li et al., 2023). Numerous studies have integrated multispectral technology with ICESat-2 to conduct bathymetric detection and inversion, leveraging both passive and active remote sensing methods. These studies have yielded significant results, primarily in island reef areas. For the bathymetry inversion in island reef areas. Cao et al. (2016) developed a high-precision bathymetry model for Ganquan Island in the South China Sea by using laser satellite data and optical imagery. This approach leverages active and passive remote sensing techniques, tailored to the specific characteristics and requirements of shallow water bathymetry. Ma et al. (2020) used ICESat-2 data and Sentinel-2 data to retrieve the bathymetry information of the Xisha Islands and Aklin Island in the South China Sea. Chu et al. (2023) considered the penetration limit bathymetry in different bands of multispectral imagery and proposed an SDB method based on spectral stratification, which was successfully applied to the long line reefs in the Nansha area of China and Buck Island in the United States Virgin Islands, improving the inversion accuracy to a certain extent. For the bathymetry inversion in polar lakes, Lin et al. (2012) used multibeam bathymetric data and Landsat TM data to invert the bathymetry of lakes in the Arctic Alaska Coastal Plain. Pope et al. (2016) also utilized the Landsat satellites with the OLI sensor, applying both the Philpot radiative transfer equation (RTE) model and a semi-empirical model based on partial in situ measurements to estimate the water volume of SGLs in western GrIS. The results were evaluated and validated using satellite stereo-derived elevation data. Similarly, Moussavi et al. (2016) also utilized the stereoscopic imaging capability of Worldview-2 data to estimate and validate the bathymetry of SGLs of the GrIS, achieving high accuracy. Williamson et al. (2018) applied this RTE-based approach to Sentinel-2 multispectral imagery. Through the synergistic use of Sentinel-2 and Landsat 8 satellites, they identified numerous drained lakes, providing algorithmic support for water depth and volume estimation. Melling et al. (2024) used Sentinel-2 data to construct RTE for different bands and validated them with toICESat-2 and ArcticDEM (Arctic Digital Elevation Model) data. Based on a rigorous adherence to physical principles, they evaluated the applicability of the RTE model to SGLs. Fricker et al. (2021) utilized ICESat-2 data to estimate the meltwater depth of the Antarctic ice sheet (AIS) and Greenland, providing a reference for the GrIS and AIS lake water depth inversion. Datta and Wouters (2021) proposed the Watta algorithm, which automatically calculates SGLs bathymetry and detects potential ice layers along tracks of the ICESat-2, focusing on the drainage situation of arctic lakes by utilizing ICESat-2 data and multispectral data. Lv et al. (2024) used the Stumpf model, combined with ICESat-2 and Sentinel-2 imagery, to invert the bathymetry of some SGLs on the GrIS from 2019 to 2023. Lutz et al. (2024) integrated ICESat-2 altimetry, in situ sonar measurements, and the RTE to establish four depth estimation methods, validated against TanDEM-X elevation models, providing a systematic methodological comparison for supraglacial lake depth and volume estimation in GrIS. Feng et al. (2025) integrated ICESat-2 and Sentinel-2 data using a multi-layer perceptron neural network for depth inversion, achieving volumetric evolution monitoring of SGLs throughout the 2022 melt season in southwestern GrIS.

**References:**

- [16] Albright, Andrea, and Craig Glennie. "Nearshore bathymetry from fusion of Sentinel-2 and ICESat-2 observations." IEEE Geoscience and Remote Sensing Letters 18.5 (2020): 900-904.
- [17] Li, Shaoyu, et al. "Satellite-derived bathymetry with sediment classification using ICESat-2 and multispectral imagery: Case studies in the South China Sea and Australia." Remote Sensing 15.4 (2023): 1026.
- [18] Cao, B., Z. G. Qiu, and B. C. Cao. "Comparison among four inverse algorithms of water depth." Journal of Geomatics Science and Technology 33.04 (2016): 388-93.
- [19] Ma, Yue, et al. "Satellite-derived bathymetry using the ICESat-2 lidar and Sentinel-2 imagery datasets." Remote Sensing of Environment 250 (2020): 112047.
- [20] Chu, Sensen, et al. "Shallow water bathymetry using remote sensing based on spectral stratification." Haiyang Xuebao 45.1 (2023): 125-137.
- [21] Lin, Zheng, Xia Li, and Jigang Qiao. "Polar lake bathymetry retrieval from remote sensing data of the arctic coastal plain in Alaska." Zhongshan Daxue Xuebao/Acta Scientiarum Natralium Universitatis Sunyatseni 51.3 (2012): 128-134.
- [22] Pope, Allen, et al. "Estimating supraglacial lake depth in West Greenland using Landsat 8 and comparison with other multispectral methods." The Cryosphere 10.1 (2016): 15-27.
- [4] Philpot, William D. "Radiative transfer in stratified waters: a single-scattering approximation for irradiance." Applied Optics 26.19 (1987): 4123-4132.
- [23] Moussavi, Mahsa S., et al. "Derivation and validation of supraglacial lake volumes on the Greenland Ice Sheet from high-resolution satellite imagery." Remote sensing of environment 183 (2016): 294-303.
- [8] Williamson, Andrew G., et al. "Dual-satellite (Sentinel-2 and Landsat 8) remote sensing of supraglacial lakes in Greenland." The Cryosphere 12.9 (2018): 3045-3065.
- [6] Melling, Laura, et al. "Evaluation of satellite methods for estimating supraglacial lake depth in southwest Greenland." The Cryosphere 18.2 (2024): 543-558.
- [24] Fricker, Helen Amanda, et al. "ICESat 2 meltwater depth estimates: application to surface melt on amery ice shelf, East Antarctica." Geophysical Research Letters 48.8 (2021): e2020GL090550.
- [25] Datta, Rajashree Tri, and Bert Wouters. "Supraglacial lake bathymetry automatically derived from ICESat-2 constraining lake depth estimates from multi-source satellite imagery." The Cryosphere Discussions 2021 (2021): 1-26.
- [26] Lv, Jinhao, et al. "Long-term satellite-derived bathymetry of Arctic supraglacial lake from ICESat-2 and Sentinel-2." The International Archives of the Photogrammetry, Remote Sensing and Spatial Information Sciences 48 (2024): 469-477.

[7] Lutz, Katrina, et al. "Assessing supraglacial lake depth using ICESat-2, Sentinel-2, TanDEM-X, and in situ sonar measurements over Northeast and Southwest Greenland." The Cryosphere 18.11 (2024): 5431-5449.

[27] Feng, Tiantian, Xinyu Ma, and Xiaomin Liu. "Volumetric evolution of supraglacial lakes in southwestern Greenland using ICESat-2 and Sentinel-2." The Cryosphere 19.7 (2025): 2635-2652.

Q25: L58: Add "to" in "validated them to ICESat-2"

**A25:** Thank you so much for your valuable comments, which have helped make the manuscript more rigorous. We have revised the manuscript accordingly.

**Q26:** L65: Given the multiple examples cited, it is inaccurate to state that the method has "rarely" been applied to SGLs. Please revise accordingly.

**A26:** Thank you so much for your valuable comments, which have helped make the manuscript more rigorous. We have removed this inaccurate statement, and the revised content is as follows:

In this study, we further extend previous methods for retrieving SGLs bathymetry. Inspired by Chu et al. (2023) on offshore islands and reefs, we applied an improved bathymetric inversion approach for SGLs that combines active and passive remote sensing.

**References:**

[20] Chu, Sensen, et al. "Shallow water bathymetry using remote sensing based on spectral stratification." Haiyang Xuebao 45.1 (2023): 125-137.

**Q27:** L74: The claim that the method supports predictions of Arctic glacier melt is overstated. The results pertain to four lakes in Southwest Greenland and should be framed accordingly.

**A27:** Thank you so much for your valuable comments, which have helped make the manuscript more rigorous. We have revised the manuscript accordingly and have explicitly clarified that the study area is located in southwestern Greenland.

Q28: L80: Remove "in the Arctic"—the location of the Greenland Ice Sheet is well known and does not need reiteration.

**A28:** Thank you so much for your valuable comments, which have helped make the manuscript more rigorous. We have revised the manuscript accordingly.

**Q29:** Figure 1: Lake D is not visible in the main Sentinel-2 background image. Consider adjusting the inset map to focus on Greenland rather than the entire Arctic to improve clarity of the study area.

**A29:** Thank you so much for your valuable comments. We have revised the manuscript accordingly. The inset map has been replaced with a section of Greenland. In addition, regarding your comment that Lake D is not visible, this is because some lakes, such as Lake D, were in a dry state at the time of imaging. As you noted, the morphology of SGLs changes rapidly, making it difficult to fully display all four lakes on the same background map. The revised inset is shown as Figure 4 (corresponding to Figure 1 in the original manuscript):

Figure 4 Revised inset map.

**Q30:** L102 & L114: Provide proper citations for the data sources (e.g., Copernicus Open Access Hub for Sentinel-2, NASA Earthdata for ICESat-2), rather than stating they were "downloaded from the internet."

**A30:** Thank you so much for your valuable comments, which have helped make the manuscript more rigorous. We have revised the manuscript accordingly. The revised contents are as follows:

The Sentinel-2 multispectral imagery data were downloaded for free from Copernicus Open Access Hub (https://dataspace.copernicus.eu/explore-data/data-collections/sentinel-data/sentinel-2).

The ICESat-2 data can be obtained freely from NASA Earthdata (https://search.earthdata.nasa.gov/).

Q31: L109–112: The description of ICESat-2 track geometry is unclear, and several terms are either unclear or possibly misapplied (e.g., orbital spacing). Rewrite this section more concisely and accurately, using standard terminology. Also, specify the ICESat-2 product used (e.g., ATL03), including version number and citation.

**A31:** Thank you so much for your valuable comments, which have helped make the manuscript more rigorous. We have revised the manuscript accordingly and provided the complete information for ICESat-2 in Table 1 of the original manuscript. The revised content is as follows:

The ICESat-2 satellite orbits at an altitude of approximately 500 km with an inclination of 92°, observing the Earth's surface between latitudes 88°S and 88°N. The platform is equipped with the Advanced Topographic Laser Altimeter System (ATLAS)

single-photon lidar and auxiliary system, which determines the distance between the spacecraft and the Earth's surface by measuring the round-trip time of photons (Markus et al., 2017). The ICESat-2/ATLAS laser emits laser pulses with a wavelength of 532 nm and a width of 1.5 ns at a frequency of 10 kHz, forming overlapping light spots along the Earth's surface with a laser footprint spacing of approximately 0.7 m (Magruder et al., 2021). The left and right points of each beam pair are approximately 90 m apart in the cross-track direction and approximately 2.5 km apart in the along-track direction. Paired tracks are approximately 3.3 km apart in the transverse track direction (Neumann et al., 2021). The detailed information of ICESat-2 used in this study is listed in Table 1. The ICESat-2 data can be obtained freely from NASA Earthdata (https://search.earthdata.nasa.gov/). Due to the rapid morphological changes of SGLs, ICESat-2 data were selected as close as possible in time to Sentinel-2 imagery. This study utilized ICESat-2 point data intercepted from Sentinel-2 data as training data, and assumed a lake water depth of 0 m at the edges of the ICESat-2 tracks. We assume that this study assumes that the depth at the intersection of the ICESat-2 track and the lake's land-water boundary is 0 m.

**References:**

- [28] Markus, Thorsten, et al. "The Ice, Cloud, and land Elevation Satellite-2 (ICESat-2): science requirements, concept, and implementation." Remote sensing of environment 190 (2017): 260-273.
- [29] Magruder, Lori, Thomas Neumann, and Nathan Kurtz. "ICESat 2 early mission synopsis and observatory performance." Earth and Space Science 8.5 (2021): e2020EA001555.
- [30] Neumann, T. A., et al. "ATLAS/ICESat-2 L2A global geolocated photon data, version 3." Boulder, Colorado USA. NASA National Snow and Ice Data Center Distributed Active Archive Center (2021).
- **Q32:** L119: Clarify that ArcticDEM strip data were used, not the mosaic. Include a brief explanation of the dataset's origin (e.g., derived from satellite stereophotogrammetry).
- **A32:** Thank you so much for your valuable comments, which have helped make the manuscript more rigorous. We have revised the manuscript accordingly. The revised contents are as follows:

It is primarily generated using satellite stereophotogrammetry, covering all land areas above 60° north latitude, with a spatial resolution of up to 2 m, and has a significant reference value for topographic research in the Arctic region.

- Q33: L121: Melling et al. (2024) is not the correct reference for ArcticDEM. Please cite the Polar Geospatial Center instead.
- **A33:** Thank you so much for your valuable comments, which have helped make the manuscript more rigorous. We have replaced the incorrect reference with the appropriate one:

Porter, Claire, et al. "ArcticDEM-Strips, Version 4.1." Harvard Dataverse https://doi. org/10.7910/DVN/C98DVS (2022).

- Q34: L125: Rather than listing acquisition dates in the text, refer to Table 1 for a clearer overview.
- **A34:** Thank you so much for your valuable comments, which have helped make the manuscript more rigorous. We have revised the manuscript accordingly by providing all the data details in Table 1 (originally) and removing the textual description. The updated Table is shown below:

Table 2. Detailed information of the datasets used in this study, the acquisition dates are highlighted in bold within the dataset names in the format yyyy/mm/dd.

| Study area | Datasets   | Data filename                                                                                    |
|------------|------------|--------------------------------------------------------------------------------------------------|
| Lake A     | Sentinel-2 | T22WEA_ 20200704 T145921                                                                  |
|            | ICESat-2   | ATL03_ 20200706 005932_01630805_005_01_gt21                                               |
|            | ArcticDEM  | $SETSM\_s2s041\_WV01\_\textbf{20200511}\_1020010094C9D900\_1020010098791800\_2m\_lsf\_seg3\_dem$ |
| Lake B     | Sentinel-2 | T22WEA_ 20200704 T145921                                                                  |
|            | ICESat-2   | ATL03_ 20200706 005932_01630805_005_01_gt31                                               |
|            | ArcticDEM  | $SETSM\_s2s041\_WV01\_\textbf{20200511}\_1020010094C9D900\_1020010098791800\_2m\_lsf\_seg3\_dem$ |
| Lake C     | Sentinel-2 | T22WEA_ 20220717 T150811                                                                  |
|            | ICESat-2   | ATL03_ 20220714 010847_03381603_006_02_gt2r                                               |
|            | ArcticDEM  | $SETSM\_s2s041\_WV01\_\textbf{20220420}\_10200100C131A200\_10200100C42D4300\_2m\_lsf\_seg1\_dem$ |
| Lake D     | Sentinel-2 | T22WEV_ 20210715 T151911                                                                  |
|            | ICESat-2   | ATL03_ 20210715 182907_03381203_006_01_gt2r                                               |
|            | ArcticDEM  | $SETSM\_s2s041\_WV02\_\textbf{20210312}\_10300100BB24B100\_10300100BBC0A100\_2m\_lsf\_seg1\_dem$ |

Q35: Table 1: Remove "Southwest Greenland Ice Sheet" from the table header—it is redundant given the context.

**A35:** Thank you so much for your valuable comments, which have helped make the manuscript more rigorous. We have revised the manuscript accordingly; the updated table has already been provided in the previous response.

Q36: L148: Revise to "unmelted ice cover \*on the lake\*" to avoid confusion with general ice sheet coverage.

**A36:** Thank you so much for your valuable comments, which have helped make the manuscript more rigorous. We have revised the manuscript accordingly.

**Q37:** L169–171: This important note about temporal alignment between ICESat-2 and Sentinel-2 data should be moved earlier in the manuscript, ideally in the data or methodology section.

**A37:** Thank you so much for your valuable comments, which have helped make the manuscript more rigorous. We have revised the manuscript accordingly and have moved this note to the data section; the relevant descriptions have already been listed in our previous response.

Q38: L222: Avoid referring to lake surface ice as "ice sheets," which has a distinct glaciological meaning.

**A38:** Thank you so much for your valuable comments, which have helped make the manuscript more rigorous. We apologize for our oversight and have revised the manuscript accordingly, as well as conducted a thorough check of the entire text.

Q39: L263–269: Consider moving this information into a table and referencing it from the main text to improve readability.

**A39:** Thank you for your suggestion. We have added figures illustrating the model differences and error analyses (Figure 1 in this document) to make the discrepancies among the models more visually intuitive. The supplementary table is provided below, including the Philpot RTE model as well as the two models originally presented in the manuscript.

Table 2. Accuracy evaluation of the three SDB models for the four lakes.

| Study  | Philpot RTE model |          |         | Lyzenga model  |          |         | Spectral stratified Lyzenga model |          |         |
|--------|-------------------|----------|---------|----------------|----------|---------|-----------------------------------|----------|---------|
| area   | $\mathbb{R}^2$    | RMSE (m) | MAE (m) | $\mathbb{R}^2$ | RMSE (m) | MAE (m) | $\mathbb{R}^2$                    | RMSE (m) | MAE (m) |
| Lake A | 0.81              | 2.94     | 2.78    | 0.96           | 0.54     | 0.43    | 0.97                              | 0.47     | 0.37    |
| Lake B | 0.85              | 3.38     | 3.26    | 0.94           | 0.67     | 0.53    | 0.95                              | 0.61     | 0.46    |
| Lake C | 0.92              | 3.73     | 3.65    | 0.97           | 0.54     | 0.40    | 0.97                              | 0.50     | 0.40    |
| Lake D | 0.81              | 1.68     | 1.41    | 0.90           | 0.76     | 0.57    | 0.91                              | 0.72     | 0.52    |

Q40: L284: The phrase "low temporal resolution" is misleading. Instead, refer to the "sparse temporal coverage" of ArcticDEM data.

**A40:** Thank you so much for your valuable comments, which have helped make the manuscript more rigorous. We have revised the manuscript accordingly. The revised contents are as follows:

Compared to the sparse temporal coverage of ArcticDEM data, the method of calculating lake volume using bathymetry information obtained by the study approach is more effective and better meets the needs of long-term, accurate monitoring of lake volume.

Finally, we would like to sincerely thank you once again for your professional and thorough review. Your comments have made our manuscript more rigorous. Admittedly, there are still aspects in the revised manuscript that can be improved, and we will continue our efforts to further advance this work.

Best regards.

---

## Author Comment (AC3)

**Responses to comments from Referee 2:**

Dear Prof. Ian Willis,

Thank you very much for your careful and thorough review of this manuscript. Your professional insights have been highly constructive for improving the paper. In your review, you provided five major comments along with many minor, detailed suggestions. Below, we have basically revised and corrected the manuscript according to your suggestions and have responded to each of your comments in detail, including revisions to the textual descriptions, redesign of relevant sections, and updates to the experimental content. In the document, your comments and suggestions are marked in black, while our responses and the revised sections of the paper based on your suggestions are marked in blue.

**Main Comments**

Q1: The paper does not adequately acknowledge all the work that has been done by many people over the last decade +, which uses a physically-based algorithm to calculate water depths on the GrIS or Antarctic ice shelves from optical imagery (MODIS, Landsat, ASTER, Sentinel-2). For example, none of the following papers is mentioned or acknowledged in this context.

. . . . .

The well used 'Philpot' approach should be acknowledged and it would also be useful to mention (or show) how the 'Philpot' equation varies from the classic and modified Lyzenga models. Of course, it would also be valuable to actually use the 'Philpot' equation (e.g. with the parameters advocated by Pope et al 2016) to compare against the 2 calibrated versions of the Lyzenga model and the ArcticDEM.

**A1:** Thank you so much for your comments. We acknowledge that the manuscript did not sufficiently recognize previous contributions in this field, which was an oversight on our part. We sincerely thank you for pointing this out and apologize for the omission. Due to space limitations, the specific references you mentioned during your review are not all listed in this document, and we kindly ask for your understanding. We have carefully revised the introduction to incorporate relevant descriptions of prior work, including those you highlighted. The revised paragraph is as follows:

For the bathymetry inversion in polar lakes, Lin et al. (2012) used multibeam bathymetric data and Landsat TM data to invert the bathymetry of lakes in the Arctic Alaska Coastal Plain. Pope et al. (2016) also utilized the Landsat satellites with the OLI sensor, applying both the Philpot radiative transfer equation (RTE) model (Philpot 1987) and a semi-empirical model based on partial in situ measurements to estimate the water volume of SGLs in western GrIS. The results were evaluated and validated using satellite stereo-derived elevation data. Similarly, Moussavi et al. (2016) also utilized the stereoscopic imaging capability of Worldview-2 data to estimate and validate the bathymetry of SGLs of the GrIS, achieving high accuracy. Williamson et al. (2018) applied this RTE-based approach to Sentinel-2 multispectral imagery. Through the synergistic use of Sentinel-2 and Landsat 8 satellites, they identified numerous drained lakes, providing algorithmic support for water depth and volume estimation. Melling et al. (2024) used Sentinel-2 data to construct RTE for different bands and validated them with ICESat-2 and ArcticDEM (Arctic Digital Elevation Model) data. Based on a rigorous adherence to physical principles, they evaluated the applicability of the RTE model to SGLs.. Fricker et al. (2021) utilized ICESat-2 data to estimate the meltwater depth of the Antarctic ice sheet (AIS) and Greenland, providing a reference for the GrIS and AIS lake water depth inversion. Datta and Wouters (2021) proposed the Watta algorithm, which automatically calculates SGLs bathymetry and detects potential ice layers along tracks of the ICESat-2, focusing on the drainage situation of arctic lakes by utilizing ICESat-2 data and multispectral data. Lv et al. (2024) used the Stumpf model, combined with ICESat-2 and Sentinel-2 imagery, to invert the bathymetry of some SGLs on the GrIS from 2019 to 2023. Lutz et al. (2024) integrated ICESat-2 altimetry, in situ sonar measurements, and the RTE to establish four depth estimation methods, validated against TanDEM-X elevation models, providing a systematic methodological comparison for supraglacial lake depth and volume estimation in GrIS. Feng et al. (2025) integrated ICESat-2 and Sentinel-2 data using a multi-layer perceptron neural network for depth inversion, achieving volumetric evolution monitoring of SGLs throughout the 2022 melt season in southwestern GrIS.

**Reference:**

- [1] Lin, Zheng, Xia Li, and Jigang Qiao. "Polar lake bathymetry retrieval from remote sensing data of the arctic coastal plain in Alaska." Zhongshan Daxue Xuebao/Acta Scientiarum Natralium Universitatis Sunyatseni 51.3 (2012): 128-134.
- [2] Pope, Allen, et al. "Estimating supraglacial lake depth in West Greenland using Landsat 8 and comparison with other multispectral methods." The Cryosphere 10.1 (2016): 15-27.
- [3] Philpot, William D. "Radiative transfer in stratified waters: a single-scattering approximation for irradiance." Applied Optics 26.19 (1987): 4123-4132.
- [4] Moussavi, Mahsa S., et al. "Derivation and validation of supraglacial lake volumes on the Greenland Ice Sheet from high-resolution satellite imagery." Remote sensing of environment 183 (2016): 294-303.
- [5] Williamson, Andrew G., et al. "Dual-satellite (Sentinel-2 and Landsat 8) remote sensing of supraglacial lakes in Greenland." The Cryosphere 12.9 (2018): 3045-3065.
- [6] Melling, Laura, et al. "Evaluation of satellite methods for estimating supraglacial lake depth in southwest Greenland." The Cryosphere 18.2 (2024): 543-558.
- [7] Fricker, Helen Amanda, et al. "ICESat 2 meltwater depth estimates: application to surface melt on amery ice shelf, East Antarctica." Geophysical Research Letters 48.8 (2021): e2020GL090550.
- [8] Datta, Rajashree Tri, and Bert Wouters. "Supraglacial lake bathymetry automatically derived from ICESat-2 constraining lake depth estimates from multi-source satellite imagery." The Cryosphere Discussions 2021 (2021): 1-26.
- [9] Lv, Jinhao, et al. "Long-term satellite-derived bathymetry of Arctic supraglacial lake from ICESat-2 and Sentinel-2." The International Archives of the Photogrammetry, Remote Sensing and Spatial Information Sciences 48 (2024): 469-477.
- [10] Lutz, Katrina, et al. "Assessing supraglacial lake depth using ICESat-2, Sentinel-2, TanDEM-X, and in situ sonar measurements over Northeast and Southwest Greenland." The Cryosphere 18.11 (2024): 5431-5449.
- [11] Feng, Tiantian, Xinyu Ma, and Xiaomin Liu. "Volumetric evolution of supraglacial lakes in southwestern Greenland using ICESat-2 and Sentinel-2." The Cryosphere 19.7 (2025): 2635-2652.

In addition, you suggested comparing the Philpot RTE model with our spectrally stratified Lyzenga model. We have adopted your valuable recommendation. Following the approaches of Lutz et al. (2024), who applied Philpot's RTE model to supraglacial lakes on the GrIS, we performed bathymetric inversion for the four lakes in this study and compared the results with those obtained using both the original and the optimized Lyzenga model.

**Q2:** The paper remains rather limited in scope. In essence it has applied two versions of an empirical model: one, the 'classic' Lyzenga model, that has already been applied to a GrIS lake by Lv et al 2024; and a modified version of it that uses different spectral bands, that has so far only been applied outside of the context of GrIS lakes. The models were calibrated using ICESat data and validated against ArcticDEM data. There appears to be quite marginal improvement in using the modified version of the model. It is a methods paper, and it doesn't tell us anything about the glaciology or hydrology of GrIS surface lakes. It would be a more compelling paper if it had gone further.

It could have remained methods based but compared also the 'Stumpf' model and the 'Philpot' model for the 4 lakes. It could have told us how transferable the model is – it appears as though the model was calibrated separately for each lake? What were the different parameter values associated with these calibrations? What happens if you apply the model calibrated for one lake to the other lakes? What errors occur? How universal is the model? What happens if you combine all the ICESat-2 data with

all the corresponding spectral data to produce an 'all lake' model? How transferable is that? The model needs testing against data that are outside of the lakes that were used for calibration.

It could also have been expanded to tell us about the hydrology of the lakes by calculating their volumes over summer seasons and between years.

**A2**: Thank you so much for your very insightful comments, and each point you made is reasonable. You have raised several key issues, and we will answer them one by one below.

**Q2** (1): The models were calibrated using ICESat data and validated against ArcticDEM data. There appears to be quite marginal improvement in using the modified version of the model. It is a methods paper, and it doesn't tell us anything about the glaciology or hydrology of GrIS surface lakes. It would be a more compelling paper if it had gone further.

**A2 (1):** We acknowledge that the manuscript lacks a direct discussion of the hydrological characteristics of lakes on the GrIS, and relevant descriptions have been appropriately added to the Introduction. The present study provides a precise remotesensing-based approach for deriving SGLs bathymetry on the GrIS, which can serve as a valuable tool for investigating more in-depth hydrological and glaciological processes. We sincerely appreciate your suggestion, which indeed represents a promising research direction, and it will be an important focus of our future work.

**Q2** (2): It could have remained methods based but compared also the 'Stumpf' model and the 'Philpot' model for the 4 lakes. It could have told us how transferable the model is – it appears as though the model was calibrated separately for each lake?

A2 (2): We have adopted your advice and added a comparative experiment using the Philpot RTE model (Philpot, 1987), which is a purely physics-based approach that differs in algorithmic principle from the model proposed in this study. Including it as a comparison is indeed meaningful. As you pointed out, the model parameters established in this study were individually calibrated for different lakes. Regarding the model's transferability, although the specific parameters in this study cannot be directly transferred, the workflow of the active and passive bathymetric model with spectral stratification is transferable, that is, dividing the optical imagery into zones based on the penetration differences among spectral bands and constraining the model parameters using a subset of in situ bathymetry data. This workflow can be applied to parameter calibration in the Stumpf model (Stumpf et al, 2003) and even integrated into other machine learning methods, as it also accounts for the physical penetration characteristics of electromagnetic waves and performs zonal modeling. This aspect ensures conceptual consistency across different active and passive bathymetry models.

**Reference:**

[3] Philpot, William D. "Radiative transfer in stratified waters: a single-scattering approximation for irradiance." Applied Optics 26.19 (1987): 4123-4132.

[12] Stumpf, Richard P., Kristine Holderied, and Mark Sinclair. "Determination of water depth with high-resolution satellite imagery over variable bottom types." Limnology and Oceanography 48.1part2 (2003): 547-556.

**Q2** (3): What were the different parameter values associated with these calibrations? What happens if you apply the model calibrated for one lake to the other lakes? What errors occur? How universal is the model?

**A2** (3): The model parameters indeed vary among different lakes. Since the parameters for each lake were individually calibrated, the mathematical relationship between water depth and spectral reflectance may not be entirely consistent across

lakes. Consequently, directly transferring the model from one lake to another would inevitably introduce errors, meaning that the current algorithm does not yet possess large-scale transferability. Nevertheless, the workflow of spectral stratification proposed in this study is generalizable, as mentioned in our response to Comment A2 (2). Thank you again for your gentle comments. We will continue to research more precise algorithms to obtain the bathymetric information of GrIS. If further explanation or modification is needed, please feel free to let us know, we will be grateful.

**Q2** (4): What happens if you combine all the ICESat-2 data with all the corresponding spectral data to produce an 'all lake' model? How transferable is that? The model needs testing against data that are outside of the lakes that were used for calibration.

**A2 (4):** This is indeed an inspiring and innovative perspective. Undoubtedly, developing an "all-lake" model based on extensive ICESat-2 data would substantially enhance the generality and transferability of the approach. In contrast, this study focuses on improving the inversion accuracy of individual lakes through an optimized spectral stratification algorithm, which necessitates separate calibration for each lake. Since the core objective of this work is accuracy improvement, the model's generality is not as strong as that of an all-lake model. We have added relevant discussion to the revised manuscript, and enhancing the model's general applicability will be an important direction for our future research.

Q2 (5): It could also have been expanded to tell us about the hydrology of the lakes by calculating their volumes over summer seasons and between years.

**A2** (5): Your comment highlights a valuable prospect for the application of our algorithm. Any optimization method must ultimately be validated through practical implementation. Regarding this point, we have previously conducted some preliminary work (Lv et al., 2024), in which the Stumpf model was used to estimate the interannual variations in the total water volume of a subset of SGLs on the GrIS over five years (Figure 1), and possible related environmental factors, such as temperature, were also discussed. This is indeed an interesting direction. The point you raised is very meaningful, and since the present study improves the model accuracy, we believe our approach could be valuable for such related research. We plan to further explore this aspect in our future work based on the method proposed in this study.

Figure 1. Volume change of a subset of SGLs on the GrIS from 2019 to 2023 (Lv et al, 2024).

**Reference:**

[9] Lv, Jinhao, et al. "Long-term satellite-derived bathymetry of Arctic supraglacial lake from ICESat-2 and Sentinel-2." The International Archives of the Photogrammetry, Remote Sensing and Spatial Information Sciences 48 (2024): 469-477.

**Q3:** The work uses the NDWI (which uses the Green and NIR bands) to separate water from ice. This is unusual as virtually all the papers I know about (see e.g., those listed above) use the variant of NDWI tailored for glacier ice and water separation that uses the blue and red bands instead of green and NIR:

NDWIce = (Blue-Red)/(Blue+Red)

Did the authors consider using this? Why did they opt for the Green and NIR bands? Could we be shown whether it makes a difference to the results?

A3: Thank you so much for your comments. We greatly appreciate your careful and professional review. We acknowledge that our consideration in this part was insufficient. The NDWI applied in our previous work (based on the green and near-infrared bands) is primarily designed for inland lakes (e.g., for water extraction over land or forested areas) and is not the optimal index for detecting water on glaciers. In glacial regions, both the green and NIR reflectance are relatively high, and their difference is much less pronounced than over soil or vegetation. This may introduce some inaccuracies when extracting glacial lake water. To address this, we have re-extracted the lake water using NDWIice, and the corresponding revisions have been made in the manuscript.

**Q4:** There is confusion throughout the paper about the definition of what the authors refer to as "a satellite-derived bathymetry (SDB) method". See lines 12-15 where it is first mentioned and defined, but then all subsequent references to it as well. Need to clarify that this is the complete method that is being used in this paper, i.e. the Lyzenga model applied to optical satellite data (Sentinel-2), calibrated using altimetry data (ICESat-2). And the paper uses two versions of the Lyzenga equation. It is not just the use of optical data alone as is sometimes implied in the manuscript.

**A4:** Thank you so much for your comments. We acknowledge that the definition and description of SDB in the manuscript were not sufficiently clear, which was an oversight on our part. We would like to clarify this section as follows:

SDB can encompass various approaches. It may involve only optical remote sensing data (e.g., Philpot's RTE method), or it may combine active and passive remote sensing techniques (such as the approach in the original manuscript, which integrates optical imagery with ICESat-2 data). Both the Lyzenga model and its optimized version used in the manuscript rely on two data sources: optical imagery and ICESat-2 altimetry. Therefore, they do not use optical imagery alone. Relevant descriptions in the manuscript have been revised to prevent any potential confusion.

**Q5:** The methodology is not clearly articulated in a consistent way throughout the paper, and parts of the methods appear in the results (see line by line comments below). A key aspect of the methodology that is not articulated clearly up front is that the ICESat-2 data are being used to calibrate the parameters of the two versions of the Lyzenga model. The equation for the 'traditional' version of the Lyzenga model and how it is applied are not explained. Does the traditional version use just one waveband from the optical data? If so which one?

**A5:** Thank you so much for your comments. We acknowledge that the description of the methods in the manuscript was not sufficiently clear. In combination with our follow-up experiments, we have revised this section to make each method more clearly defined. In the original manuscript, both the traditional Lyzenga model and the optimized Lyzenga model use the same spectral bands, namely the blue and green bands. The difference between them is that the optimized Lyzenga model employs a zonal modeling approach, using different model parameters for each partition to partially address potential errors arising from the varying penetration capabilities of electromagnetic waves at different wavelengths.

Line by line comments

**Q6:** 15 'verify' => 'validate' [as 'validate' is used subsequently in the paper].

**A6:** Thank you so much for your valuable comments, which have helped make the manuscript more rigorous. We have revised the manuscript accordingly.

Q7: 17. say you're using 'time stamped' DEM strips here.

**A7:** Thank you so much for your valuable comments, which have helped make the manuscript more rigorous. We have revised the manuscript accordingly.

**Q8:** 19-21. Do you need a sentence before this one explaining what you did to enable you to make this statement? How did you prove it is 'scalable' and what do you mean by that? I'm not convinced you've proved it's scalable, have you?

**A8:** Thank you so much for your valuable comments, which have helped make the manuscript more rigorous. We have revised the manuscript accordingly. We have removed the description of "scalable" from the abstract.

**Q9:** 26. Is Luthje et al a good reference here? Does it explicitly consider 'ecological' issues?

**A9:** Thank you so much for your valuable comments. The study by Luthje et al. does not explicitly address ecological issues; we have revised the manuscript accordingly.

**Q10:** 28-30. Beckmann and Winkelmann, 2023 do not consider lakes in their paper as implied. Clarify what you mean by this sentence and use an appropriate ref.

**A10:** Thank you so much for your valuable comment, which has helped make the manuscript more rigorous. The sentence was intended to convey that Arctic glacier melt has negative impacts on the global ecosystem. We have accordingly revised the references in the manuscript.

Q11: 32. You can't sail ships on ice sheet surface lakes so delete this! Your 3 refs all relate to lidar. Could you quote studies using bathymetry from small boats, e.g.:

- 1. Box, J. E., & Ski, K. (2007). Remote sounding of Greenland supraglacial melt lakes: implications for subglacial hydraulics. Journal of Glaciology, 53(181), 257–265. https://doi.org/10.3189/172756507782202883. Cambridge University Press & Assessment
- 2. Tedesco, M., & Steiner, N. (2011). In-situ multispectral and bathymetric measurements over a supraglacial lake in western Greenland using a remotely controlled watercraft. The Cryosphere, 5, 445–452. https://doi.org/10.5194/tc-5-445-2011. Copernicus TC

Both papers include in-situ depth sounding (Box & Ski with a raft + depth sounder; Tedesco & Steiner with a remotely controlled boat equipped with GPS and sonar).

**A11:** Thank you so much for your valuable comments. We sincerely apologize for our oversight. The relevant statements in the manuscript have been revised to accurately reflect the facts. The corresponding parts in the manuscript have been updated accordingly as follows:

Accurately estimating the volume of these lakes requires detailed bathymetry data, which is particularly challenging to obtain due to the harsh climatic conditions in the Arctic. Although conventional bathymetric methods, including bathymetric airborne lidar and shipborne sonar, have achieved high levels of maturity and accuracy, They have also been successfully applied over the GrIS (Box and Ski, 2007; Tedesco and Steiner, 2011); however, due to limitations in cost and timeliness (Li et al., 2022; Qi et al., 2022; Qi et al., 2024), these methods cannot meet the monitoring demands of rapidly changing SGLs on GrIS.

**Reference:**

- [13] Box, Jason E., and Kathleen Ski. "Remote sounding of Greenland supraglacial melt lakes: implications for subglacial hydraulics." Journal of glaciology 53.181 (2007): 257-265.
- [14] Tedesco, Marco, and N. Steiner. "In-situ multispectral and bathymetric measurements over a supraglacial lake in western Greenland using a remotely controlled watercraft." The Cryosphere 5.2 (2011): 445-452.
- [15] Li, Shaoyu, et al. "Bathymetric LiDAR and multibeam echo-sounding data registration methodology employing a point cloud model." Applied Ocean Research 123 (2022): 103147.
- [16] Qi, Chao, et al. "A method to decompose airborne LiDAR bathymetric waveform in very shallow waters combining deconvolution with curve fitting." IEEE Geoscience and Remote Sensing Letters 19 (2022): 1-5.
- [17] Qi, Chao, et al. Analysis and correction in the airborne LiDAR bathymetric error caused by the effect of seafloor topography slope, National Remote Sensing Bulletin, (2024)26, 2642-2654.
- **Q12:** 32. You say 'in polar regions' here. But I'd suggest focussing on the GrIS earlier in your introduction and stop referencing Arctic / Polar regions after that.
- **A12:** Thank you so much for your valuable comments, which have helped make the manuscript more rigorous. We have revised the manuscript accordingly and carefully checked the entire text to ensure consistency and accuracy.
- Q13: 41. These references are not on the GrIS so make that clear in the sentence. You could reference other papers relevant for the GrIS and AIS.
- **A13**: Thank you so much for your valuable comments, which have helped make the manuscript more rigorous. We have updated the relevant references accordingly.
- Q14: 42-3. "...integrated multispectral technology with ICESat-2 to conduct bathymetric detection and inversion, leveraging both active and passive remote sensing..." Here and elsewhere in the paper (e.g. 47-8, 64) it would be useful to stick to one order and not switch the order you mention these two types of satellite data. So here you could write "integrated multispectral technology with ICESat-2 to conduct bathymetric detection and inversion, leveraging both passive and active remote sensing". It's a trivial point but it makes for a clearer, more logical read.
- **A14:** Thank you for your suggestion, which has helped make the manuscript more rigorous. We have revised the manuscript accordingly.
- Q15: 44. You refer to 'the mainland' but you need to be clearer in this section that the methods you're using were first applied in settings outside the GrIS. Tell us what mainland here as it reads like you're talking about mainland Greenland!

**A15:** Thank you for your suggestion, which has helped make the manuscript more rigorous. We have revised the manuscript accordingly. To avoid confusion, the phrase "far away from the mainland" has been removed.

Q16: 46. "...leverages active and passive remote sensing techniques..." You told us that 3 lines up so delete. More useful to tell us what sensors.

**A16:** Thank you so much for your valuable comments, which have helped make the manuscript more rigorous. We have revised the manuscript accordingly, making some reductions while retaining more concise and effective information. The revised contents are as follows:

Cao et al. (2016) developed a high-precision bathymetry model for Ganquan Island by combining laser bathymetry and WorldView-2 imagery, leveraging both active and passive approaches.

**Reference:**

[18] Cao, B., Z. G. Qiu, and B. C. Cao. "Comparison among four inverse algorithms of water depth." Journal of Geomatics Science and Technology 33.04 (2016): 388-93.

Q17: 50. Full stop after 'Bermuda."

**A17:** Thank you so much for your valuable comments, which have helped make the manuscript more rigorous. We have revised the manuscript accordingly.

Q18: 50. 'proposed' is the wrong word. Do you mean 'divided'? Or 'classified'?

**A18:** Thank you so much for your valuable comments, which have helped make the manuscript more rigorous. We have revised the manuscript accordingly and removed this citation, as including too many examples from non-polar regions was deemed unnecessary.

Q19: 54. You say "improving the inversion accuracy" but compared to what?

**A19:** Thank you so much for your valuable comments, which have helped make the manuscript more rigorous. We have revised the manuscript accordingly and removed this citation, as including too many examples from non-polar regions was deemed unnecessary.

**Q20:** 56-7. Before the sentence spanning these lines, I'd specify that you're focussing on surface lakes on ice masses. Note it's not just for 'Arctic regions' as you include Fricker et als work on Antarctic surface lakes in your references.

**A20:** Thank you so much for your valuable comments, which have helped make the manuscript more rigorous. We have replaced 'Arctic regions' with 'polar lakes'.

**Q21:** 61. "Watta algorithm". This is the 3rd algorithm that's been introduced now with no details. It'd be more important to tell us the basis of this (and earlier) algorithms not what Datta and Wouters called it. I assume they called it this because it is an amalgamation of their names, but this is not really crucial information here.

**A21:** Thank you so much for your valuable comments, which have helped make the manuscript more rigorous. We have revised the manuscript accordingly. The revised content is as follows:

Datta and Wouters (2021) proposed the Watta algorithm, which automatically calculates SGLs bathymetry and detects potential ice layers along tracks of the ICESat-2, focusing on the drainage situation of arctic lakes by utilizing ICESat-2 data and multispectral data.

**Reference:**

[7] Datta, Rajashree Tri, and Bert Wouters. "Supraglacial lake bathymetry automatically derived from ICESat-2 constraining lake depth estimates from multi-source satellite imagery." The Cryosphere Discussions 2021 (2021): 1-26.

**Q22:** 61. YOU say 'arctic lakes' but it was just for GrIS lakes. Having already focussed down on the GrIS in your review, I'd stick to referencing work relevant to the GrIS here and not keep mentioning 'Arctic' or 'Polar regions' etc.

**A22:** Thank you so much for your valuable comments, which have helped make the manuscript more rigorous. We have revised the manuscript accordingly and have made clearer distinctions in the manuscript, no longer conflating the GrIS with the broader Arctic Polar region.

Q23: 63. Delete 'some' and delete 'from 2019 to 2023' This latter not relevant.

**A23:** Thank you so much for your valuable comments, which have helped make the manuscript more rigorous. We have revised the manuscript accordingly.

Q24:65. 'polar' See my comment above for line 61.

**A24:** Thank you so much for your valuable comments, which have helped make the manuscript more rigorous. We have revised the manuscript accordingly.

**Q25:**65-7. Is this just the Lv et al (2024) paper that's relevant here? If so say so. But you should also acknowledge Mousavi et al, Pope et al, Williamson et al, Melling et al, etc. who did consider different bands using the 'Philpot' algorithm.

**A25:** Thank you so much for your valuable comments, which have helped make the manuscript more rigorous. We have revised the manuscript accordingly and have fully cited the previous studies you mentioned; the revisions have already been provided in our response to the previous comment.

Q26:68 Suggest "...Chu et al. (2023) to offshore islands..."

**A26:** Thank you so much for your valuable comments, which have helped make the manuscript more rigorous. We have revised the manuscript accordingly.

**Q27:** 68-70. So you're building on Lv et al 2024 which so far is the only paper to apply what you're calling the SDB method to GrIS lakes, but you're adding spectral stratification as used by Chu et al in a different context? Could state this more clearly

A27: Thank you so much for your valuable comments, which have helped make the manuscript more rigorous. We have provided more detailed descriptions in the manuscript to more clearly illustrate the relationships among these methods. The revised content is as follows:

In this study, we further extend previous methods for retrieving SGLs bathymetry. Inspired by Chu et al. (2023) on offshore islands and reefs, we applied an improved bathymetric inversion approach for SGLs that combines active and passive remote

sensing.

**Reference:**

[19] Chu, Sensen, et al. "Shallow water bathymetry using remote sensing based on spectral stratification." Haiyang Xuebao

45.1 (2023): 125-137.

**Q28:** 72-3. Ok so you're 'calibrating' using ICESat-2 and validating with ArcticDEM. This could be expressed more clearly.

A28: Thank you so much for your valuable comments, which have helped make the manuscript more rigorous. The section

you mentioned indeed required clarification, and we have revised the manuscript accordingly. The revised content is as follows:

Specifically, ICESat-2 lidar data were integrated with Sentinel-2 multispectral imagery, taking into account the varying

penetration abilities of different spectral bands (i.e., red, green, blue, and near-infrared). The Sentinel-2 imagery was divided

into multiple spectral layers using the Otsu algorithm to construct a spectral stratification-based Lyzenga model. In this

framework, ICESat-2 lake bottom photons were used as training samples to build semi-empirical models for each spectral

layer, thereby improving the accuracy of SGL bathymetric estimation. For comparison, we also applied the Philpot RTE model

and the traditional Lyzenga model without spectral stratification optimization to the same study area. All results were validated

against high-resolution ArcticDEM data.

**Q29:** 74 'Arctic' => 'GriS'

A29: Thank you so much for your valuable comments, which have helped make the manuscript more rigorous. We have

revised the manuscript accordingly.

Q30: 74. "... offers effective technical support for predicting Arctic glacier melt and global climate change." This is too grand.

I'd delete this. You can't do this based on your work.

A30: Thank you so much for your valuable comments, which have helped make the manuscript more rigorous. The previous

statement is overstated. The manuscript has been revised accordingly; the revised contents are as follows:

This study provides a new and more accurate approach for monitoring the volumes of SGLs and offers methodological

guidance for SGL bathymetry research.

Q31: 75-7. Suggest delete.

10

**A31:** Thank you so much for your valuable comments, which have helped make the manuscript more rigorous. We have revised the manuscript accordingly.

Q32: 80-2. Suggest delete "in the Arctic, the second-largest ice sheet in the world, surpassed only by the Antarctic Ice Sheet. However, the GrIS is more fragile and sensitive to temperature changes than the Antarctic Ice Sheet (Robinson et al., 2012)." Everybody knows the first statement and the 2nd is a bit vague.

**A32:** Thank you so much for your valuable comments, which have helped make the manuscript more rigorous. We have revised the manuscript accordingly.

Q33: 85. 'aimed => 'aims' and "verify" => 'validate'

**A33:** Thank you so much for your valuable comments, which have helped make the manuscript more rigorous. We have revised the manuscript accordingly.

Q34: 86 "bathymetry data ... were used" [data are plural]

**A34:** Thank you so much for your valuable comments, which have helped make the manuscript more rigorous. We have revised the manuscript accordingly.

Q35: 91. Its blue green red yellow respectively

**A35:** Thank you so much for your valuable comments, which have helped make the manuscript more rigorous. We have revised the manuscript accordingly.

Q36: Fig 1. Your Arctic inset is rather ugly. I suggest use another inset - just for the GrIS - and remove the words 'study area' from the map.

**A36:** Thank you so much for your comments. We have revised the manuscript accordingly. The inset has been changed to show Greenland, and the updated study area map is presented below:

Figure 2. Revised SGLs insets

Q37: 100-1. Suggest change to "Sentinel-2 imagery was obtained for lakes A and B on 4 July 2020, lake C on 17 July 2022, and lake D on 15 July 2021."

**A37:** Thank you so much for your valuable comments, which have helped make the manuscript more rigorous. We have revised the manuscript accordingly. The related statements have been removed, and the acquisition dates are now directly presented in Table 1 to avoid redundant descriptions. The revised table is as follows:

Table 1. Detailed information of the datasets used in this study, the acquisition dates are highlighted in bold within the dataset names in the format yyyy/mm/dd.

| Study area | Datasets   | Data filename                                                                                    |  |
|------------|------------|--------------------------------------------------------------------------------------------------|--|
| Lake A     | Sentinel-2 | T22WEA_ 20200704 T145921                                                                  |  |
|            | ICESat-2   | ATL03_ 20200706 005932_01630805_005_01_gt2l                                               |  |
|            | ArcticDEM  | $SETSM\_s2s041\_WV01\_\textbf{20200511}\_1020010094C9D900\_1020010098791800\_2m\_lsf\_seg3\_dem$ |  |
| Lake B     | Sentinel-2 | T22WEA_ 20200704 T145921                                                                  |  |
|            | ICESat-2   | ATL03_ 20200706 005932_01630805_005_01_gt3l                                               |  |
|            | ArcticDEM  | $SETSM\_s2s041\_WV01\_\textbf{20200511}\_1020010094C9D900\_1020010098791800\_2m\_lsf\_seg3\_dem$ |  |
| Lake C     | Sentinel-2 | T22WEA_ 20220717 T150811                                                                  |  |
|            | ICESat-2   | ATL03_ 20220714 010847_03381603_006_02_gt2r                                               |  |
|            | ArcticDEM  | $SETSM\_s2s041\_WV01\_\textbf{20220420}\_10200100C131A200\_10200100C42D4300\_2m\_lsf\_seg1\_dem$ |  |
| Lake D     | Sentinel-2 | T22WEV_ 20210715 T151911                                                                  |  |
|            | ICESat-2   | ATL03_ 20210715 182907_03381203_006_01_gt2r                                               |  |
|            | ArcticDEM  | SETSM_s2s041_WV02_ 20210312 _10300100BB24B100_10300100BBC0A100_2m_lsf_seg1_dem            |  |

Q38: 102. 'can be' => 'was' [say what you did not what is possible to do]

**A38:** Thank you so much for your valuable comments, which have helped make the manuscript more rigorous. We have revised the manuscript accordingly and carefully checked the entire text.

Q39: 110-11. Suggest "The left and right points of each beam pair are approximately 90 m apart in the transverse track direction and about 2.5 km apart in the along-track direction."

**A39:** Thank you so much for your valuable comments, which have helped make the manuscript more rigorous. We have revised the manuscript accordingly.

**Q40:** 112-13. Suggest: "The ICESat-2 data used in this study were acquired on 6 July 2020 (lakes A and B), 15 July 2021 (lake C), and 14 July 2022 (lake D)."

**A40:** Thank you so much for your valuable comments, which have helped make the manuscript more rigorous. We have revised the manuscript accordingly. The related statements have been removed, and the acquisition dates are now directly presented in Table 1

Q41: 115 "data, and"

**A41:** Thank you so much for your valuable comments, which have helped make the manuscript more rigorous. We have revised the manuscript accordingly.

**Q42:** 115-16. "Please note that this study assumes" => "We assume"

**A42:** Thank you so much for your valuable comments, which have helped make the manuscript more rigorous. We have revised the manuscript accordingly.

**Q43:** 118 "can be" => "data were"

**A43:** Thank you so much for your valuable comments, which have helped make the manuscript more rigorous. We have revised the manuscript accordingly.

Q44: 119 delete "and has the characteristics of a large coverage area and high spatial resolution."

**A44:** Thank you so much for your valuable comments, which have helped make the manuscript more rigorous. We have revised the manuscript accordingly.

Q45: 121. Delete "high-resolution Arctic digital elevation model ArcticDEM"

**A45:** Thank you so much for your valuable comments, which have helped make the manuscript more rigorous. We have revised the manuscript accordingly.

Q46: 122-4. Ambiguous. Tell the reader what you did - what data you used.

A46: Thank you so much for your valuable comments. We have revised the manuscript accordingly as follows:

In this study, the most recent version (s2s041) of the ArcticDEM strip was used to validate the accuracy of the bathymetry and volume estimated results.

Q47: 124. Delete "It should be pointed out that"

**A47:** Thank you so much for your valuable comments, which have helped make the manuscript more rigorous. We have revised the manuscript accordingly.

**Q48:** 124-5. Explain why these dates. Presumably the first data to cf. with your derived bathymetry along the ICESat-2 line for lakes A and B, 2nd for bathymetry of lake C, 3rd for lake D?

**A48:** Thank you so much for your valuable comments. We removed this part from the text description and instead presented the data correspondence more intuitively in Table 1.

**Q49:** 130-1. You say "To address the challenges and the limitations of traditional bathymetric methods, which do not consider the varying penetration of electromagnetic waves of different wavelengths into water..." But this is not true as many previous studies using the 'Philpot' method have done this.

**A49:** Thank you so much for your valuable comments. We apologize for our oversight and have revised the manuscript accordingly. Previous studies based on the Philpot RTE method have indeed considered the differences in water penetration among spectral bands. What we intended to emphasize here is the approach of dividing the entire lake into different zones according to variations in electromagnetic wave penetration, and performing bathymetric inversion separately in each zone. This process is implemented by combining ICESat-2 bathymetric data with the zonal Lyzenga model. We have thoroughly revised this part accordingly as follows:

To improve the accuracy of SDB for SGLs of GrIS, this study compares three approaches: the physically-based RTE model, the traditional Lyzenga model, and a novel spectrally stratified optimized Lyzenga model. While the RTE model relies on physical parameterization of water column properties, and the traditional Lyzenga model combines multiple spectral bands empirically, while both approaches incorporate spectral information through wavelength-dependent attenuation coefficients, they generally do not stratify the retrieval model based on the varying effective depth ranges of different spectral bands. Inspired by the spectral stratification method applied to shallow coral reefs by Chu et al. (2023), this study adapts and extends this approach to SGLs. The proposed method combines ICESat-2 data with Sentinel-2 multispectral imagery. Using the Otsu algorithm (Otsu, 1975), spectral stratification is performed based on radiance differences at various water depths across different bands. The stratified spectral layers are then combined with ICESat-2 bathymetric data to construct optimized Lyzenga models for each spectral layer.

**Reference:**

- [19] Chu, Sensen, et al. "Shallow water bathymetry using remote sensing based on spectral stratification." Haiyang Xuebao 45.1 (2023): 125-137.
- [20] Otsu, Nobuyuki. "A threshold selection method from gray-level histograms." Automatica 11.285-296 (1975): 23-27.

**Q50:** Figure 2. This implies you're producing two versions of the Lyzenga model - traditional and spectral stratified. And you'll compare them both against Arctic DEM data? This was not mentioned as part of your methodology earlier in the abstract or in your brief overview of the methodology for the work on lines 68-73. It should be clearer earlier that this is part of your work.

**A50:** Thank you so much for your valuable comments, which have helped make the manuscript more rigorous. We have revised the manuscript accordingly. We have clearly stated in both the abstract and the introduction that two methods were used. In addition, we have added relevant descriptions of the RTE method. The revised flowchart is as follows:

Figure 3. Revised flowchart, with the SDB model used in this study highlighted in yellow in the illustration.

Q51: 140. Delete "provided by the ESA" and say "...data are..."

**A51** Thank you so much for your valuable comments, which have helped make the manuscript more rigorous. We have revised the manuscript accordingly.

Q52: 142. '... can be processed...' Again, tell us what you did not what can be done.

**A52:** Thank you so much for your valuable comments, which have helped make the manuscript more rigorous. We have revised the manuscript accordingly.

**Q53:** 144-6. Why did you not use the version for separating water from ice? NDWI\_ice? there's another variant of NDWI tailored for glacier ice detection that uses the blue and red bands instead of green and NIR.

That formulation is:

NDWIice =(Blue-Red)/(Blue+Red)

- Blue: reflectance in the blue band (~0.45 μm, e.g., Sentinel-2 Band 2)
- Red: reflectance in the red band (~0.65 μm, e.g., Sentinel-2 Band 4)

**A53:** Thank you so much for your valuable comment. We acknowledge that our consideration in this part was insufficient. The NDWI version using the Green and NIR bands is typically applied for water extraction in vegetation or soil environments rather than in glacial settings. Therefore, in this study, we have replaced it with NDWI for water extraction and revised the subsequent experimental sections accordingly. The revised text is as follows:

The Sentinel-2 data are divided into L1C and L2A levels. The L1C level data product is a geometric precision correction radiographic product that has not undergone radiometric correction. L2A products are products that undergo radiation correction processing based on L1C. The water column was extracted from multispectral imagery using water-land separation methods, i.e., the Normalized Difference Water Index (*NDWI*), specifically its variant adapted for water extraction in ice–snow covered environments, Eq. (4) (originally Equation 1 in the manuscript) (McFeeters, 1996; Yang et al., 2012), combined with threshold-based grayscale segmentation.

$$NDWI_{lce} = \frac{Blue - Red}{Blue + Red} \tag{1}$$

Where *Blue* represents the reflectance at the blue band (corresponding to Sentinel-2 Band 2) and *Red* represents the reflectance at the red band (corresponding to Sentinel-2 Band 4).

The image was divided into water and non-water parts using the  $NDWI_{Ice}$ , with the non-water portion—including lake ice—applied as a mask to extract the open-water body. As the multispectral imagery contained partially unmelted ice on the lake, this masking process effectively removed ice-covered areas and ensured accuracy during the water column extraction.

**References:**

- [21] McFeeters, Stuart K. "The use of the Normalized Difference Water Index (NDWI) in the delineation of open water features." International journal of remote sensing 17.7 (1996): 1425-1432.
- [22] Yang, Kang, and Laurence C. Smith. "Supraglacial streams on the Greenland Ice Sheet delineated from combined spectral–shape information in high-resolution satellite imagery." IEEE Geoscience and Remote Sensing Letters 10.4 (2012): 801-805.

Q54: 151. 'data are'

**A54:** Thank you so much for your valuable comments, which have helped make the manuscript more rigorous. We have revised the manuscript accordingly.

Q55: 152. Delete "In this study"

**A55:** Thank you so much for your valuable comments, which have helped make the manuscript more rigorous. We have revised the manuscript accordingly.

Q56: 154 and 155 "should say 'the four'

**A56:** Thank you so much for your valuable comments, which have helped make the manuscript more rigorous. We have revised the manuscript accordingly.

**Q57:** 155 "Due to the fact that" => 'Because"

**A57:** Thank you so much for your valuable comments, which have helped make the manuscript more rigorous. We have revised the manuscript accordingly.

**Q58:** 157-8. "Finally, more accurate bathymetric photons were obtained for constructing a bathymetry inversion model." I don't understand this sentence and it splits up the two either side of it that are related. I suggest delete this sentence (or clarify what it means if it's important).

**A58:** Thank you so much for your valuable comments. We have revised the manuscript accordingly. We agree that this sentence may be confusing. Our intended meaning was that the ICESat-2 data, after refraction correction, can more accurately represent the underwater topography. To avoid ambiguity, this sentence has been deleted.

Q59: 169. Delete "It should be noticed that"

**A59:** Thank you so much for your valuable comments, which have helped make the manuscript more rigorous. We have revised the manuscript accordingly.

**Q60:** 176-7. Suggest "This study applied the spectral stratification method using the Otsu algorithm, which automatically determines thresholds without requiring input parameters (Otsu, 1975)."

**A60:** Thank you so much for your valuable comments, which have helped make the manuscript more rigorous. We have revised the manuscript accordingly.

**Q61:** 177-8. This is not a sentence. The main clause is missing a finite verb. Right now, "multispectral images of water stratified into four layers" is written as if it's a complete statement, but it lacks a clear subject performing an action. Should it say '...water were stratified..." ? But 'stratified into 4 layers' implies lake depth layers, but you don't mean this - you're giving 4 wavebands. This is confusing.

**A61:** Thank you so much for your valuable comments, which have helped make the manuscript more rigorous. We have revised the manuscript accordingly.

Q62: 179. Could delete 'layer'

**A62:** Thank you so much for your valuable comments, which have helped make the manuscript more rigorous. We have revised the manuscript accordingly.

Q63: 189. It'd be useful to show the equation for this "traditional Lyzenga model" too.

**A63:** Thank you so much for your valuable comments. We have revised the manuscript accordingly and briefly derived its relationship with the Philpot RTE model to make the logic more coherent. The revised content is as follows:

**3.2.2 Traditional Lyzenga model**

Both Lyzenga multi-band logarithmic linear model (Lyzenga, 1978, 1985) and Philpot RTE models exploit the exponential attenuation of light in water following Beer-Lambert's law, with Philpot's RTE approach providing explicit physical parameterization of bottom reflectance and water column properties that were empirically combined in Lyzenga's earlier formulation. Unlike the purely theoretical RTE approach, which derives water depth only from optical imagery, this method introduces empirical constraints using a limited number of measured depth values to estimate model parameters. In this formulation, the bottom reflectance ( $A_d$ ) and the diffuse attenuation coefficient function (g) are treated as constants, where  $a_0=ln(A_d-R_\infty)/g$ ,  $a_1=-1/g$ . At this stage, the model equation is written as:

$$Z = a_0 + a_1 \left( R_\omega - R_\infty \right) \tag{2}$$

By integrating the spectral reflectance information from multiple bands, the following expression can be obtained:

$$Z = a_0 + a_i \sum_{i=1}^{n} \ln \left[ R_{\omega}(\lambda_i) - R_{\infty}(\lambda_i) \right]$$
(3)

This represents the commonly used Lyzenga model. In this study, ICESat-2 bathymetric points are incorporated to constrain the empirical parameters of the Lyzenga model, and the parameter estimation is performed using the Levenberg–Marquardt (LM) algorithm.

**Reference:**

[23] Lyzenga, David R. "Passive remote sensing techniques for mapping water depth and bottom features." Applied optics 17.3 (1978): 379-383.

[24] Lyzenga, David R. "Shallow-water bathymetry using combined lidar and passive multispectral scanner data." International journal of remote sensing 6.1 (1985): 115-125.

Q64: 191. You say "...combining the traditional Lyzenga model..." But with what? This is unclear.

**A64:** Thank you so much for your valuable comments, which have helped make the manuscript more rigorous. We have revised the manuscript accordingly. What we intended to express here is that different spectral layers are combined with ICESat-2 data to establish zonal Lyzenga models for different regions. The revised content is as follows:

By leveraging the varying penetration abilities of electromagnetic waves in water, the multispectral images were segmented into different spectral layers, and a spectral-stratified Lyzenga model was established by integrating ICESat-2 data with water reflectance in each layer.

**Q65:** 192-3. "the near-infrared layer, the red layer, and the green layer were combined for processing. In other words, Arctic SGLs were divided into green and blue layers." This is contradictory. Did you use the NIR and Red or just the Green? Explain more precisely what you did.

**A65:** Thank you so much for your valuable comments, which have helped make the manuscript more rigorous. We have revised the manuscript accordingly. Since the threshold-segmented areas of the NIR and Red bands are very small, the available

ICESat-2 control points are extremely limited. Therefore, in this study, the NIR, Red, and Green spectral layers were combined and collectively referred to as the "green layer".

**Q66:** 197-8. So what values do these parameters take? You haven't really explicitly stated you're fitting these equations using the ICESat-2 data to derive Z, then you're using these calibrated equations to derive bathymetry for the whole lake. Please in your methods explain the calibration process explicitly. In the results it would be useful to know what values these parameters take for the two equations for the 4 lakes. It'd also be useful to know what the equation for the original Lyzenga model is, how that works, and what the parameter values are for that and how they vary between lakes.

**A66:** Thank you so much for your valuable comments. Accordingly, we have revised the manuscript to specify that the parameter estimation method used in this study is the Levenberg–Marquardt (LM) algorithm. In addition, all model parameters applied in this study will be listed in a newly added appendix or supplement to help readers better understand the proposed algorithm.

**Q67:** 210 'performed' => 'performs'

**A67:** Thank you so much for your valuable comments, which have helped make the manuscript more rigorous. We have revised the manuscript accordingly.

**Q68:** 211 'was' => 'is'

**A68:** Thank you so much for your valuable comments, which have helped make the manuscript more rigorous. We have revised the manuscript accordingly.

**Q69:** 213. It's only here that it becomes apparent that you're constructing 2 models. The original Lyzenga and a spectral stratified version of Lyzenga? This has not been clear throughout your methodology sections so far. For example, you do not mention this in Section 3.

**A69:** Thank you so much for your valuable comments, which have helped make the manuscript more rigorous. We have added a description of the traditional Lyzenga model.

Q70: 216-22. This should all have been stated above in methods. Not here. Start the results section with the results!

**A70:** Thank you so much for your valuable comments, which have helped make the manuscript more rigorous. We have revised the manuscript accordingly and removed these irrelevant descriptions.

Q71: 229. Where have you used the active (ICESat-2) data to derive Fig 5? This was not adequately explained.

**A71:** The ICESat-2 data were used as control points to constrain the construction of the Lyzenga model, and this has been clarified in the revised manuscript.

**Q72:** 231. You say "...underscoring the reliability and feasibility of the spectral stratified model". But you cannot conclude this until you've compared with the Arctic DEM. Delete this.

**A72:** Thank you so much for your valuable comments, which have helped make the manuscript more rigorous. We have revised the manuscript accordingly.

Q73: 232. You don't need any of these phrases ending with the word 'that', e.g. 'It should be noted that...'. Just delete this and all such phrases.

**A73:** Thank you so much for your valuable comments, which have helped make the manuscript more rigorous. We have revised the manuscript accordingly.

**Q74:** 232-3. You say " the ArcticDEM data only contains spatial information of the lake bottom, and lacks water surface elevation information when obtaining bathymetry benchmark data". Is that true? Esp as you use early season strips. How do you know the lakes were empty at that time rather than containing water and frozen over?

**A74:** Thank you so much for your valuable comments, which are well justified. The ICESat-2 and ArcticDEM profiles used in this study show good agreement. ICESat-2 is capable of detecting lake bottom information, and if the lakes contained water covered by ice, the DEM would display a flat surface. However, this is not the case with the ArcticDEM data used here. Therefore, it can be reasonably assumed that the ArcticDEM represents empty lake basins, consistent with the assumptions in Melling et al. (2024) and related studies.

**Q75:** 236-7. You say "However, the sediment in the lakes of the experimental area primarily consists of bedrock, a type of material that remains stable and does not undergo significant changes over short periods." This makes no sense to me at all.

**A75:** Thank you so much for your valuable comments. We are sorry that the statement in this section lacked sufficient scientific basis and theoretical support, and we have supplemented and clarified the description in the revised manuscript, as follows:

The GrIS is covered by ice, and both the ice surface and supraglacial lakes evolve, making short-term morphological changes unavoidable. However, as noted by Echelmeyer et al. (1991), many large supraglacial lakes remain fixed in space because their surface depressions are dynamically supported by irregularities in the underlying bedrock. This bedrock control limits large-scale spatial shifts within short periods, although minor variations are inevitable. Consequently, some discrepancies between ICESat-2-derived bathymetry and ArcticDEM data collected several months apart are expected. Nevertheless, the overall consistency between the two datasets remains strong, as also demonstrated in the experiment by Melling et al. (2024). These differences fall within an acceptable range and do not compromise the reliability of ArcticDEM as a high-quality reference dataset.

**Reference:**

[5] Melling, Laura, et al. "Evaluation of satellite methods for estimating supraglacial lake depth in southwest Greenland." The Cryosphere 18.2 (2024): 543-558.

[25] Echelmeyer, Keith, T. S. Clarke, and Will D. Harrison. "Surficial glaciology of jakobshavns isbræ, West Greenland: Part I. Surface morphology." Journal of Glaciology 37.127 (1991): 368-382.

In addition, to clarify this point, we have excerpted relevant descriptions from these two papers:

'Due to the sparse temporal sampling of ArcticDEM and the need to resolve empty basins, the DEMs are not temporally concurrent with the ICESat-2 and Sentinel-2 data. As a result, the smallest period between the ArcticDEM and ICESat-2 acquisition dates was approximately 2 months (Lake 4), and the largest period was approximately 11 months (Lake 5) (Table A1). As the location and shape of supraglacial lakes are determined by bedrock topography (Echelmeyer et al., 1991), we assume there should be little change in the bathymetry of the lake basins between the data acquisition dates (see Sect. 3.1 for further details).' Melling et al. (2024)

'Repeated photogrammetry and surveying shows that many large lakes remain fixed in space and that they are not advected along with the moving ice. This seems to imply that the larger surface depressions are tied to bedrock irregularities and are thus dynamically supported.' Echelmeyer et al. (1991)

**Q76:** 239. Having read to here, I'm not convinced you needed any of the previous section 4.1 on the qualitative analysis. Consider deleting it as the quantitative analysis is what is needed.

**A76:** Thank you so much for your valuable comments, which have helped make the manuscript more rigorous. We have added spatial maps of the inverted bathymetric errors for the lakes in this section to support the qualitative analysis. The analysis figures are shown as follows:

To further evaluate the performance differences among the models, residuals between the bathymetry derived from ArcticDEM and that obtained from each model were calculated for qualitative comparison. The spatial distribution of the residuals is illustrated in Figure 4 (corresponding to Figure 6 in the original manuscript). In addition, since the ArcticDEM data only contains spatial information of the lake bottom, and lacks water surface elevation information when obtaining bathymetry benchmark data. Therefore, in this study, the edge position of the ArcticDEM lake was determined using ICESat-2 data, with this elevation serving as the water surface elevation.

Figure 4. Differences between the water results derived from the three SDB models and those obtained from ArcticDEM. Red areas indicate overestimation, while blue areas indicate underestimation.

**Q77:** 242. "spectral stratified Lyzenga model" . Check entire paper and refer to this model in the same way throughout. This is quite clear here but you've not so far ever referred to it in this way.

**A77:** Thank you so much for your valuable comments, which have helped make the manuscript more rigorous. We have revised the manuscript accordingly.

Q78: 246. 'visually demonstrate' => "illustrate"

**A78:** Thank you so much for your valuable comments, which have helped make the manuscript more rigorous. We have revised the manuscript accordingly.

Q79: 247-9. This sentence is obvious and could be deleted.

**A79:** Thank you so much for your valuable comments, which have helped make the manuscript more rigorous. We have revised the manuscript accordingly.

**Q80:** 249-50. Do you mean to refer to Fig 7 here as it doesn't show ArcticDEM validation. Do you mean Fig 8? If so refer to Fig 7 earlier.

**A80:** Thank you so much for your valuable comments, which have helped make the manuscript more rigorous. After considering the revised experimental content and the overall structure of the manuscript, we believe that the contents described in Figures 7 and 8 (originally L246–L260) are no longer necessary to retain. This is because we have supplemented spatial error maps for all lake model inversion results relative to ArcticDEM, and all results have been quantitatively evaluated for accuracy. These additions fully cover the information previously presented in those figures.

**Q81:** 251. " spectral stratified model" See my comment for line 242. Try to refer to this model consistently throughout the paper.

**A81:** Thank you so much for your valuable comments, which have helped make the manuscript more rigorous. We have revised the manuscript accordingly.

**Q82:** 256-7. You say "...are primarily concentrated in the 2-6 m range, with more pronounced differences at the transitions between different spectral layers. This is hard to see, esp. the last point as you don't mark on where the different spectral layer data were used to construct the bathymetry.

**A82:** Thank you so much for your valuable comments, which have helped make the manuscript more rigorous. We have revised the manuscript accordingly. We agree that this part makes it difficult to discern the differences clearly. Moreover, after reviewing the context before and after this section, we found that the description does not contribute significantly to the overall logic, so it has been removed. Regarding your comment that "different spectral layer data were used to construct the bathymetry," we would like to clarify that in this study, spectral stratified Lyzenga models were established for different spectral layers, and this has been clearly described in the revised methodology section.

**Q83:** Fig 8. I'd assumed this would be a subset of Fig 6a, but for the data in the 2 sub-areas shown in Fig 7. But it's 'density'. Why are you showing this? It needs explaining.

**A83:** Thank you so much for your valuable comments. This part has been removed from the revised manuscript, as the differences between lakes can already be clearly seen from the difference maps and scatter plots, making the additional description redundant.

**Q84:** 276-7. Delete this sentence from here. This belongs in the Abstract and the start of the Conclusion.

**A84:** Thank you so much for your valuable comments, which have helped make the manuscript more rigorous. We have revised the manuscript accordingly.

**Q85:** 278. You say "volume changes" but this implies you're going to determine volume change through time, i.e. applied your final model over more time periods. But you don't do this, which is a shame. Clarify what you mean by 'volume changes' here.

**A85:** Thank you so much for your valuable comments. We acknowledge that our study focused solely on volume estimation and did not include an analysis of volume variation, and apologize for the inappropriate wording. Accordingly, the term "change" has been removed from the revised manuscript, and long-term variation analysis will be addressed in our future research.

**Q86:** Figure 9. I'm not fully convinced these add anything although they're nice visually. Suggest put in Supplementary Materials. But what would be more useful to know is what are the volumes of the 4 lakes derived by the traditional Lyzenga method and from the Arctic DEM? How do the 3 estimates compare? Which overestimates vs. underestimates cf. others?

**A86:** Thank you so much for your valuable comments. We have calculated the lake volumes derived from each model and compared them with those obtained from ArcticDEM. The corresponding analysis has been added to the Discussion section of the manuscript as follows:

**5.1 Evaluation of the lake volume estimation**

Based on the inverted bathymetry, the lake volume can be readily obtained by integrating the product of pixel area and water depth. In this study, the volumes derived from the Philpot RTE model, the Lyzenga model, and the spectral stratified Lyzenga model were calculated and compared with those obtained from ArcticDEM, as summarized in Table 2.

Table 2. Volume estimation based on the three SDB model and comparison with ArcticDEM.

| Study area | SDB model                         | ArcticDEM volume (m³) | Estimated volume (m 3 ) | Relative difference (m 3 ) |
|------------|-----------------------------------|-----------------------|------------------------------------|---------------------------------------|
| Lake A     | Philpot RTE model                 |                       | 15 700 000                         | +6 690 000 (+74%)                     |
|            | Lyzenga model                     | 9 010 000             | 9 620 000                          | +610 000 (+7%)                        |
|            | Spectral stratified Lyzenga model |                       | 9 480 000                          | +470 000 (+5%)                        |
| Lake B     | Philpot RTE model                 |                       | 5 870 000                          | +2 750 000 (+88%)                     |
|            | Lyzenga model                     | 3 120 000             | 3 360 000                          | +229 000 (+8%)                        |
|            | Spectral stratified Lyzenga model |                       | 3 350 000                          | +235 000 (+7%)                        |
| Lake C     | Philpot RTE model                 |                       | 6 560 000                          | +2 720 000 (+71%)                     |
|            | Lyzenga model                     | 3 840 000             | 3 650 000                          | -189 000 (-5%)                        |
|            | Spectral stratified Lyzenga model |                       | 3 710 000                          | -127 000 (-3%)                        |
| Lake D     | Philpot RTE model                 | 16 200 000            | 26 100 000                         | +9 900 000 (+61%)                     |

| Lyzenga model                     | 16 800 000 | +600 000 (+4%) |
|-----------------------------------|------------|----------------|
| Spectral stratified Lyzenga model | 16 500 000 | +300 000 (+2%) |

As shown in Table 2, the Philpot RTE model exhibited a consistent overestimation of lake volume across all four sites, ranging from 61% to 88%, which is considerably higher than that obtained from the Lyzenga and spectral stratified Lyzenga models. The latter two produced comparable results, with only minor differences observed for some lakes. Although the overall volume estimations derived from the spectral stratified Lyzenga model are close to those from the conventional Lyzenga model, the stratified approach offers a more physically interpretable framework by considering the spectral heterogeneity within the water body. In particular, this method allows each optical subset to be characterized by a separate empirical relationship, which may better reflect the inherent variations in water optical properties. While the improvement in total volume estimation is limited in this study, the model design provides a conceptually sound extension of the traditional Lyzenga model, and its potential advantages may become more evident in environments with stronger spatial or spectral variability.

**Q87:** 289-90. You say "While the spectral stratification-based method enhances the accuracy of bathymetric inversion compared to traditional approaches, such as the Lyzenga model..." But you can't say this. You can't generalise. You can only refer to the traditional Lyzenga model here, not ALL 'traditional approaches' by which I assume you also mean the Stumpf model? As that is the only other one you mentioned in your Intro. Did you ever consider applying this one and comparing it? And of course it'd have been valuable and interesting to have applied the 'Philpot' method too and evaluated that.

**A87:** Thank you so much for your valuable comments. The point you raised is indeed crucial. It is true that we previously generalized all traditional methods and claimed that the spectral stratification algorithm improved upon them, which was inaccurate. Accordingly, we have revised both the text and the experimental design, adding the Philpot RTE model for bathymetry inversion as a reference. The experimental results are shown in Figures 5 and 6.

Figure 5. Bathymetry results using the supplementary Philpot RTE method.

Figure 6. Accuracy validation of the supplementary Philpot RTE method.

**Q88:** 291. 'amount' => 'number'

**A88:** Thank you so much for your valuable comments, which have helped make the manuscript more rigorous. We have revised the manuscript accordingly.

**Q89:** 291. What do you mean 'strip-shaped data'? Do you mean " orbital track"?

**A89:** Thank you so much for your valuable comments. The term we described indeed refers to the orbital track, and "strip-shaped data" in the manuscript has been replaced with "orbital track data."

**Q90:** 291-2. You say there were not enough training sample data. But each lake seems well sampled along the tracks according to Fig 3. What was the total number of sampling points for each lake? Are you saying you need more? Is that really the case?

**A90:** Thank you so much for your valuable comments. We acknowledge that our original statement was unclear and may have caused confusion. What we intended to convey is not that the total number of ICESat-2 points is insufficient, but rather that their spatial distribution leads to uneven coverage of training points. ICESat-2 provides sufficient points only along its tracks, leaving other areas of the lakes often underrepresented. After spectral stratification, some spectral layers, such as the red band layer, cover very small areas, resulting in even fewer matching ICESat-2 points. Therefore, these layers had to be combined with other spectral layers; this is indeed a major challenge. The revised content is as follows:

Due to the distribution characteristics of ICESat-2 orbital tracks, the available bathymetric samples were not spatially well distributed but were confined along the ICESat-2 orbital tracks. This limitation restricted the full training of the model. Consequently, in this study, the NIR, Red, and Green bands had to be combined for model construction. If additional spectral stratification based on the red band could be introduced, the detection accuracy might be further improved.

**Q91:** 292. You say the lack of training data "...hindered further improvement in model accuracy" which is poorly phrased but more importantly, you can't conclude that. Perhaps extra data would not have improved accuracy (when cf. ArcticDEM). It MAY have improved the estimation of the model parameters and thereby led to greater accuracy, but you don't know this.

**A91:** Thank you so much for your valuable comments. We agree that this statement lacked rigor; we have removed the relevant description to avoid any possible misunderstanding.

**Q92:** 293-4. You say "...where bathymetric variations are minimal, the impact of spectral penetration differences is limited, resulting in only marginal improvements in accuracy." Did you point out such areas? Looking at Fig 6 I do not see greater 'improvements' at high depths cf. shallow depths.

**A92:** Thank you so much for your valuable comments. Your comment is well taken. The differences between the Spectral Stratified Lyzenga model and the original Lyzenga model are indeed minimal, making it difficult to highlight such regions. In the original manuscript, Figures 7 and 8 were used to visualize relatively noticeable areas and assess their accuracy; however, they still appear very similar visually. Therefore, we have revised the relevant description to make it more rigorous, as follows:

Additionally, because the SGLs are characterized by clear water, shallow depth, and stable substrates, the reflectance of the lake water shows no significant overall variation across different regions. Therefore, the impact of spectral penetration differences is limited, resulting in only marginal changes and improvements in accuracy between the Lyzenga model and the spectral stratified model, as shown in Figures 5 and 6.

Q93: 294. 'Second,...'

**A93:** Thank you so much for your valuable comments, which have helped make the manuscript more rigorous. We have revised the manuscript accordingly.

At last, we would like to once again express our heartfelt thanks for your professional and detailed review. Your comments are of great importance to the optimization of this manuscript. As you rightly pointed out, there are indeed many shortcomings in this work. We have made every effort to revise it, and those aspects that we were not able to fully address will become important objectives of our future work.

Best regards.